# Myosin-VIIa is expressed in multiple isoforms and essential for tensioning the hair cell mechanotransduction complex

Sihan Li[1,2], Andrew Mecca[3], Jeewoo Kim[1], Giusy A. Caprara[3], Elizabeth L. Wagner[1,2], Ting-Ting Du[1], Leonid Petrov [4], Wenhao Xu[5], Runjia Cui[6], Ivan T. Rebustini[6], Bechara Kachar[6], Anthony W. Peng [3,7✉] & Jung-Bum Shin[1,2,7✉]

Mutations in myosin-VIIa (MYO7A) cause Usher syndrome type 1, characterized by combined deafness and blindness. MYO7A is proposed to function as a motor that tensions the hair cell mechanotransduction (MET) complex, but conclusive evidence is lacking. Here we report that multiple MYO7A isoforms are expressed in the mouse cochlea. In mice with a specific deletion of the canonical isoform (*Myo7a-ΔC* mouse), MYO7A is severely diminished in inner hair cells (IHCs), while expression in outer hair cells is affected tonotopically. IHCs of *Myo7a-ΔC* mice undergo normal development, but exhibit reduced resting open probability and slowed onset of MET currents, consistent with MYO7A's proposed role in tensioning the tip link. Mature IHCs of *Myo7a-ΔC* mice degenerate over time, giving rise to progressive hearing loss. Taken together, our study reveals an unexpected isoform diversity of MYO7A expression in the cochlea and highlights MYO7A's essential role in tensioning the hair cell MET complex.

[1] Department of Neuroscience, University of Virginia, Charlottesville, VA, USA. [2] Department of Biochemistry and Molecular Genetics, University of Virginia, Charlottesville, VA, USA. [3] Department of Physiology and Biophysics, University of Colorado Anschutz Medical Campus, Aurora, CO, USA. [4] Department of Mathematics, University of Virginia, Charlottesville, VA, USA. [5] Genetically Engineered Murine Model (GEMM) Core, University of Virginia, Charlottesville, VA, USA. [6] National Institute for Deafness and Communications Disorders, National Institute of Health, Bethesda, MD, USA. [7] These authors contributed equally: Anthony W. Peng, Jung-Bum Shin. ✉email: anthony.peng@cuanshutz.edu; js2ee@virginia.edu

Hair cells, the sensory receptors of the auditory and vestibular system, transduce mechanical stimuli from sound and head movement into an electrochemical signal[1–3]. The mechanosensory hair bundle, consisting of an array of elongated microvilli called stereocilia, harbors the mechano-electrical transduction (MET) complex. The tip link, an extracellular filament bridging two adjacent stereocilia, connects the MET channel complex at its lower end with the side of the stereocilium in the next tallest row. Deflection of the hair bundle increases tension in the tip links, which opens MET channels at its lower end, leading to the influx of cations (primarily $K^+$ and $Ca^{2+}$) and subsequent hair cell depolarization. Over the past 4 decades, human and mouse genetics studies in combination with targeted candidate approaches have revealed the molecular correlates of principal functional features of the MET complex[4]: the tip link is now accepted to consist of a heterotetramer of CDH23 and PCDH15[5–8]. The long elusive MET channel complex harbors the proteins TMC1/2, TMIE, LHFPL5, and CIB2[9–15].

Critically, the upper end of the tip link is hypothesized to house a tension-generating element that biases the MET channels to be at an optimally sensitive open probability. According to the prevailing model, the tension generator anchors and pulls the tip link upward along the F-actin core (illustrated in Fig. 1a). The resulting tensioning of the tip link sets the resting open probability ($P_o$) of the MET channel. Furthermore, the sliding of the motor down the F-actin filaments may provide the mechanistic basis for the so-called slow adaptation process in hair cells[16–18]. The tip-link motor is thus a critical determinant of the sensitivity and dynamic range of hair cell MET. Several myosin motors have been proposed to fulfill this function, including myosin-Ic (MYO1C)[17,19] and MYO7A[20]. The most recent studies on localization[21] and motor activity[22] advanced MYO7A as a likely candidate for the molecular motor. The adaptor proteins USH1G (sans) and USH1C (harmonin)[21,23–25] presumably provide the scaffold by which the F-actin-bound motor connects to the upper end of the tip link. Consistent with this notion, mutations in harmonin were shown to affect the resting $P_o$ of the MET channel[24]. However, functional evidence establishing MYO7A as the tension generator has been challenging to obtain, because hair bundle development is affected in knockout and knockdown mouse models of MYO7A, precluding isolated loss-of-function analyses.

In this study, we made the unexpected discovery that inner hair cells (IHCs) and outer hair cells (OHCs) preferentially express

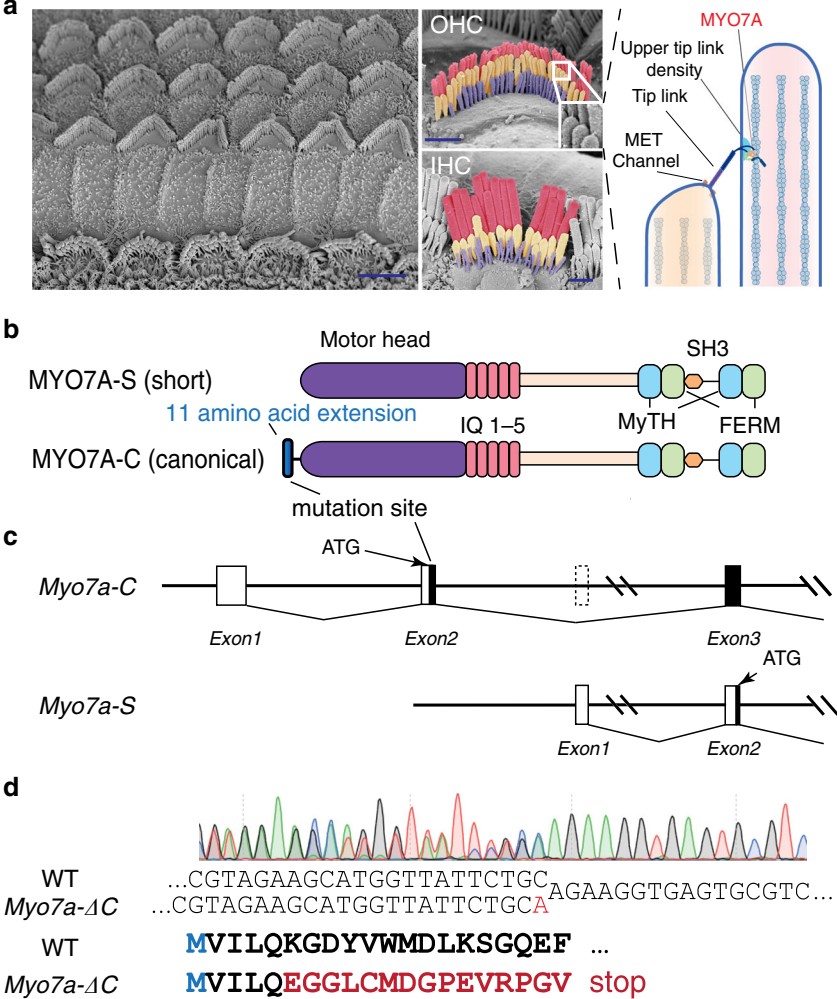

**Fig. 1 Specific deletion of the canonical MYO7A-C isoform. a** Scanning electron micrographs depicting IHCs and OHCs in the mouse organ of Corti, and a schematic illustration of the proposed function of MYO7A at the upper tip-link density of the mechanotransduction complex (scale bars: 5 μm in overview SEM, and 1 μm in OHC and IHC SEMs). **b**, **c** Graphical illustration of the canonical and the presumed short MYO7A protein isoforms (MYO7A-C and MYO7A-S, respectively) and corresponding genomic structures. The site of the deleterious mutation in the *Myo7a-ΔC* mouse is indicated. **d** Sanger sequencing result of the heterozygous *Myo7a-ΔC* founder mouse. An adenine insertion (shown in red) introduces a deleterious mutation in the reading frame of *Myo7a-C*. Sequencing was performed from the reverse direction.

different isoforms of the candidate tip-link motor protein MYO7A. In genetically engineered mice with a specific deletion of the canonical MYO7A isoform (*Myo7a-ΔC* mouse), MYO7A levels are severely reduced in IHCs, but the hair bundle develops normally. This mouse model provided an avenue to specifically test the role of MYO7A in hair cell MET function. The changes in MET current properties, hair bundle morphology and hearing function in the *Myo7a-ΔC* mouse are in support for a direct role of MYO7A in tensioning the hair cell MET complex.

## Results

**Multiple isoforms of MYO7A are expressed in the cochlea**. Analysis of genomic sequence databases revealed the existence of multiple MYO7A isoforms. Two such isoforms are generated by alternative transcription and translation start sites (Fig. 1b, c). The translation start site (ATG) of the canonical isoform (MYO7A-C) is located in exon 2, while the predicted start site of a shorter isoform from database (MYO7A-S) is located two exons downstream. Despite the 10-kb-long intervening genomic sequence between the two start sites, MYO7A-C is distinguished from MYO7A-S only by an 11-amino acid (aa) N-terminal extension to the myosin head domain (Fig. 1b).

To investigate the expression and functional relevance of these two MYO7A isoforms in hair cells, we generated a mouse line in which the canonical isoform was specifically deleted (*Myo7a-ΔC* mouse). Using CRISPR/Cas9 genome editing[26], a truncating frameshift mutation (1 bp insertion) in exon 2 (ten bases downstream of the start codon) of the mouse *Myo7a* gene was introduced (Fig. 1b–d). *Myo7a-ΔC* mice had no overt behavioral phenotype and all parts of the inner ear developed normally.

**Myo7a-C deletion primarily affects MYO7A in cochlear IHCs**. We examined the spatiotemporal expression of MYO7A in *Myo7a-ΔC* mice. Strikingly, MYO7A immunoreactivity was strongly diminished in IHCs, while expression in OHCs was not overtly affected (Fig. 2a, c). Despite the significant reduction in MYO7A levels, IHCs in the *Myo7a-ΔC* mice had WT-like bundle morphology at postnatal day 5 (P5). These observations were in contrast to the severe morphological defects present in the hair bundles of *Myo7a full KO* mice (Fig. 2c), which we generated independently by introducing a deleterious mutation in exon 24. Hair bundle morphology was analyzed in more detail in P7 hair cells: the lengths of the longest and second row IHC stereocilia were measured in volume-rendered phalloidin fluorescence confocal images. No significant differences between *Myo7a-ΔC* and WT counterparts were observed ($p = 0.706$ and $p = 0.936$ for first and second row, respectively) (Fig. 2e). Additional analyses of stereocilia morphologies using stereopairs of scanning electron micrographs (SEMs) using a modified version of previously described methods[27,28] (described in detail in Supplementary Fig. 5), also revealed no significant differences between *Myo7a-ΔC* and WT IHCs ($p = 0.563$ and $p = 0.364$ for first and second row, respectively) and OHCs ($p = 0.248$, $p = 0.755$, $p = 0.64$ for first, second, and third row, respectively) (Fig. 2f, g). We suggest that the seemingly normal hair bundle development in cochlear IHCs is due to the residual and redundant expression of unaffected MYO7A isoforms, presumably MYO7A-S, in the *Myo7a-ΔC* mice. In addition, in the vestibular utricle of *Myo7a-ΔC* mice, MYO7A expression in all hair cells, as determined by relative immunofluorescence (IF) to MYO6, was reduced by 63% ($1.35 \pm 0.19$ in WT; $n = 9$ utricles compared with $0.52 \pm 0.11$ in *Myo7a-ΔC*; $n = 9$ utricles; $p < 1e-4$), without cell-type specific distinction (Fig. 2b).

A more detailed analysis of the remaining cellular expression pattern of MYO7A in the *Myo7a-ΔC* mice uncovered tonotopic

differences in OHCs: MYO7A levels in OHCs of *Myo7a-ΔC* mice were similar to WT levels in basal regions of the cochlea, but decreased in the middle and apical turns by ~30% and 52% ($p < 1e-3$ and $<1e-4$), respectively, (Fig. 3a, b). The quantification was conducted by normalizing the MYO7A immunoreactivity to MYO6 immunoreactivity, which was invariable between *Myo7a-ΔC* and WT mice and along the tonotopic axis (Supplementary Fig. 1). We therefore inferred that MYO7A-C is predominantly expressed in all IHCs and in a tonotopic gradient in OHCs, decreasing from the apex toward the base of the cochlea. The presumed alternative isoform MYO7A-S, which cannot be specifically detected due to sequence overlap with MYO7A-C (Fig. 1b, c), is hypothesized to be expressed weakly in IHCs (constituting ~15% of overall MYO7A expression level) and in OHCs in a tonotopic gradient that runs counter to that of MYO7A-C.

Attempts to generate a MYO7A-C-specific antibody failed due to the low antigenicity of the 11-aa N-terminal peptide. To obtain direct evidence for the cellular expression of MYO7A-C, we generated the *HA-Myo7a-C* KI mouse line, by knocking-in a HA-tag immediately after the MYO7A-C start codon (Fig. 3c). Consistent with the phenotype of the *Myo7a-ΔC* mouse, HA-MYO7A-C immunoreactivity was strong in all IHCs, and in OHCs, decreased toward the base of the cochlea (Fig. 3d). The same expression pattern was observed in mature (P30) *HA-Myo7a-C* KI mice (Supplementary Fig. 2), demonstrating that this expression is not a transient phenomenon during development.

Next, we revisited the activity of a previously characterized *Myo7a* promoter that is widely used to drive transgene expression in hair cells and is derived from the genomic region upstream of *Myo7a-C*, including the untranslated exon 1 (Fig. 3e)[29,30]. Contrary to the general assumption, but consistent with the hypothesized expression pattern of MYO7A-C, *Myo7a* promoter-driven actin-GFP fluorescence was restricted to IHCs and apical OHCs (Fig. 3f). Occasional actin-GFP fluorescence was observed in OHCs of the middle turns as well. We therefore conclude that the expression driven by the widely used *Myo7a* promoter replicates the expression pattern of the canonical MYO7A-C isoform only.

**Reduced MYO7A at the upper tip-link density (UTLD) and stereocilia base of *Myo7a-ΔC* IHCs**. MYO7A was reported to localize to several sites within the hair cell in addition to the diffuse staining throughout the cytosol, including the stereocilia base and the UTLD. The localization at the UTLD is believed to be essential for MYO7A's proposed function as the tip-link motor[21], while at the stereocilia base, MYO7A is involved in establishing the ankle-link complex[31–35]. Using a previously characterized antibody[21], we examined the localization of MYO7A at the UTLD and the stereocilia base. At the UTLD of *Myo7a-ΔC* IHCs, MYO7A immunoreactivity was detected but reduced by 68% at P7 ($p < 1e-3$) and 58% at P16 ($p < 1e-3$) compared with WT. At the stereocilia base, MYO7A signal was reduced by 81% at P7 ($p < 1e-3$) and 69% at P16 ($p < 1e-3$) (Fig. 4a–d). Compared with the 85% ($p < 1e-4$) reduction of cytosolic MYO7A (Fig. 3b, for middle region IHCs), the degree of MYO7A reduction was therefore comparable at the stereocilia base, but less pronounced at the UTLD, suggestive of MYO7A sequestration through high-affinity binding sites at the UTLD.

In our hands, the localization of MYO7A at the UTLD could only be visualized with the antibody developed by Grati et al.[21], while commercially available MYO7A antibodies failed to reproduce this staining. The *HA-Myo7a-C KI* mouse model represented an opportunity to independently validate MYO7A's localization at the UTLD, which is of high significance for its

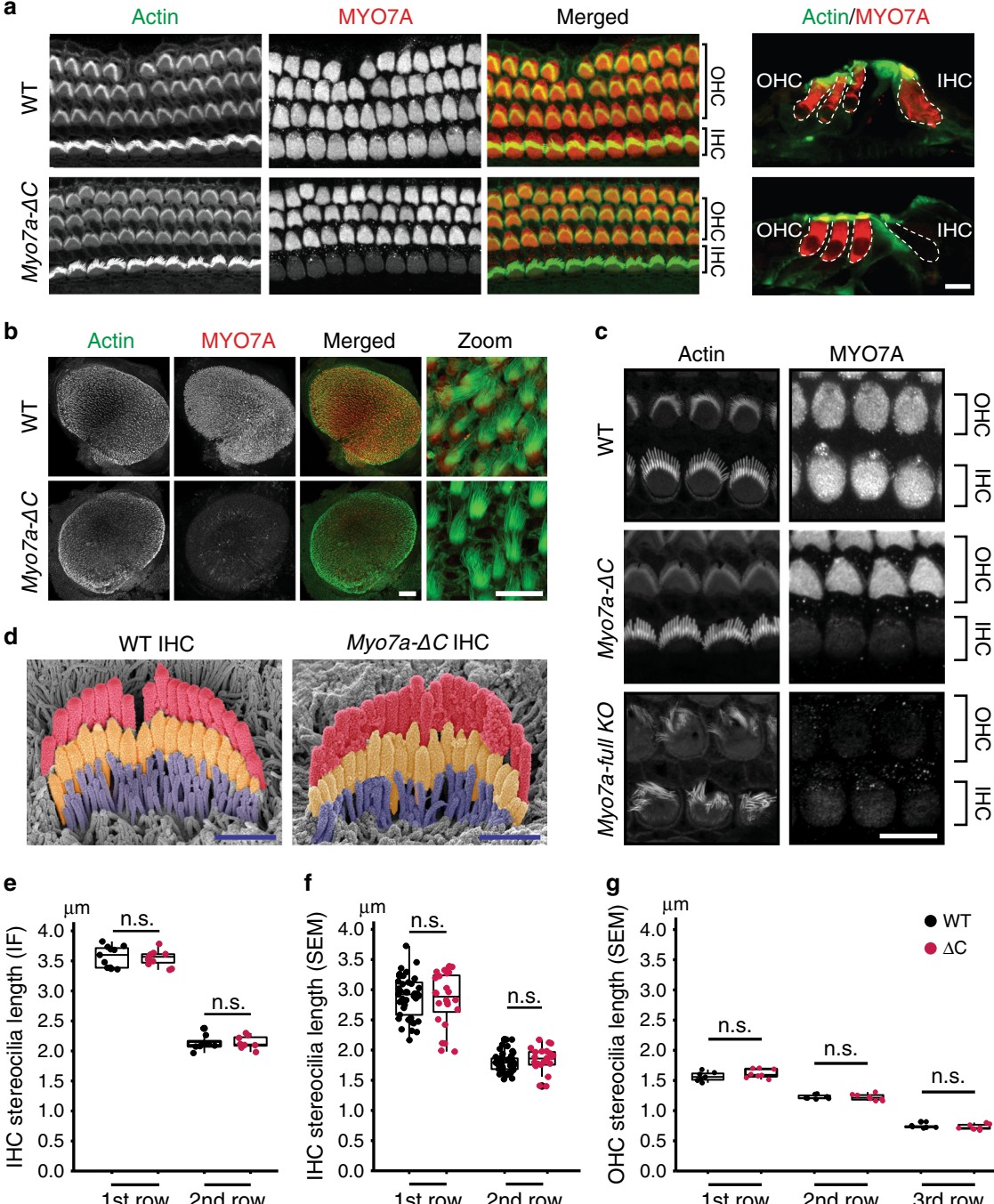

**Fig. 2 _Myo7a-C_ deletion primarily affects MYO7A expression in IHCs and utricle hair cells. a** MYO7A immunoreactivity in WT and _Myo7a-ΔC_ organ of Corti (P5), counterstained with phalloidin (F-actin). MYO7A is predominantly decreased in the IHCs (quantified in Fig. 3b) (scale bars: 10 μm). **b** MYO7A is significantly reduced in all utricle hair cells (by 63%, _p_ < 1e−4, two-tailed, unpaired _t_-test). (scale bars: 100 μm left, 10 μm right). **c** MYO7A immuno- and phalloidin reactivity in IHCs and OHCs in WT, _Myo7a-ΔC_, and _Myo7a full KO_ mice (scale bar: 10 μm). Although MYO7A levels are depleted in _Myo7a-ΔC_ IHCs, hair bundle morphology is not affected. **d** Representative SEMs of IHCs of P7 WT and _Myo7a-ΔC_ (scale bars: 1 μm). **e** No difference in stereocilia length in P7 IHCs between WT and _Myo7a-ΔC_ mice by phalloidin immunofluorescence (IF) analysis (using Imaris 3D module). Length of first row stereocilia (mean per cell ± SD): WT (3.57 ± 0.18); _Myo7a-ΔC_ (3.54 ± 0.13), _p_ = 0.706. Analyzed numbers (stereocilia, cells, animals): WT (159; 7; 4), _Myo7a-ΔC_ (127; 8; 3). Second row (mean per cell ± SD): WT (2.13 ± 0.13), _Myo7a-ΔC_ (2.14 ± 0.11), _p_ = 0.936. Analyzed numbers (stereocilia, cells, animals): WT(89; 8; 4), _Myo7a-ΔC_ (95; 8; 3) (n = number of cells). **f** Quantification of stereocilia length in P7 IHCs by SEM found no significant differences between WT and _Myo7a-ΔC_ mice. Length of first row stereocilia (mean per cell ± SD): WT (2.88 ± 0.35), _Myo7a-ΔC_ (2.83 ± 0.45), _p_ = 0.664. Second row: WT (1.80 ± 0.16), _Myo7a-ΔC_ (1.84 ± 0.19), _p_ = 0.363. Analyzed numbers (stereocilia, cells, animals): WT (141; 41; 10), _Myo7a-ΔC_ (123; 24; 7). **g** Quantification of stereocilia length in P7 OHCs by SEM found no significant differences between WT and _Myo7a-ΔC_ mice. Length of first row stereocilia (mean per cell ± SD): WT (1.57 ± 0.077), _Myo7a-ΔC_ (1.61 ± 0.074), _p_ = 0.23. Length of second row (mean per cell ± SD): WT (1.23 ± 0.036), _Myo7a-ΔC_ (1.22 ± 0.052), _p_ = 0.80. Length of third row (mean per cell ± SD): WT (0.74 ± 0.036), _Myo7a-ΔC_ (0.73 ± 0.045), _p_ = 0.64. Analyzed numbers (stereocilia, cells, animals): WT(73; 7; 3), _Myo7a-ΔC_ (70; 8; 3). _p_ values of **e**–**g** were derived from two-tailed, unpaired _t_-tests. Boxplots indicate medians, 25th and 75th percentiles as box limits, and minima and maxima as whiskers, respectively. _N_ for statistical analysis in **e**–**g** was number of cells. Source data are provided in the Source Data file.

proposed role as the tip-link motor. IF imaging using a commercially available HA antibody resolved the membrane-adjacent localization of HA-MYO7A-C, as well as MYO7A enrichment at the predicted site of the UTLD and the stereocilia base. In a subset of stereocilia, MYO7A signal was also observed at the stereocilia tips (Fig. 4e). The significance of the tip localization is unknown and remains to be investigated in future studies.

**Harmonin and ADGRV1 are not decreased in *Myo7a-ΔC* IHCs.** A tripartite complex consisting of MYO7A, sans, and harmonin was shown to constitute the major protein components of the UTLD. Harmonin, the most abundant protein of the UTLD[21], is required for UTLD formation and its hypofunction affects tip-link tension[24,36]. Furthermore, a previous study showed that its localization to the UTLD is dependent on MYO7A[36]. We therefore investigated whether the reduced levels of MYO7A in *Myo7a-ΔC* IHCs affects the localization of harmonin. Consistent with previous studies[21], harmonin localization in WT IHC and OHC stereocilia was readily detected at the presumed position of the UTLD (Fig. 5a, b). Interestingly, despite a 68% reduction of MYO7A levels at the UTLD, harmonin immunoreactivity was not reduced, but rather slightly elevated in *Myo7a-ΔC* IHCs (Fig. 5a–c). Furthermore, the average number of harmonin puncta per cell was not significantly different between *Myo7a-ΔC* IHCs and WT counterparts ($11.1 \pm 3.16$ in WT and $12.3 \pm 2.62$ in *Myo7a-ΔC*, $t$-test $p = 0.271$). This is in contrast to the severe mislocalization of harmonin (formation of aggregates in the cuticular plate) that was observed in *Myo7a full KO* hair cells (Fig. 5a), consistent with a previous study[36]. A previous study also demonstrated that the localization of the ankle-link component ADGRV1 (also known as VLGR1, GPR98, MASS1) was disturbed in *Myo7a null Shaker-1* mice[31]. Immunolocalization of ADGRV1 using a previously described antibody[37] demonstrated that ADGRV1 at the stereocilia base was not significantly altered in *Myo7a-ΔC* IHCs (Fig. 5d, e). We conclude that the residual level of MYO7A in *Myo7a-ΔC* IHCs is sufficient to mediate the targeting of harmonin and ADGRV1 to the UTLD and the ankle-link complex, respectively.

**Reduced resting $P_o$ and slowed MET currents in *Myo7a-ΔC* IHCs.** Despite an 85% reduction of total MYO7A levels, IHCs in *Myo7a-ΔC* mice exhibited WT-like hair bundle morphology in the first postnatal week as observed by light and scanning electron microscopy (Fig. 2d, e). Furthermore, harmonin localization was not affected in *Myo7a-ΔC* IHCs (Fig. 5b, c), suggesting that the reduction of MYO7A did not affect the overall composition and integrity of the tip-link motor complex at the UTLD. The *Myo7a-ΔC* IHCs therefore provided a powerful tool to directly and specifically test the role of MYO7A in hair cell MET, without the confounding effects arising from defects in hair bundle development or MET complex integrity.

To examine the potential contribution of MYO7A-C to MET characteristics, we conducted whole-cell voltage-clamp recordings in the middle-apical turn of WT and *Myo7a-ΔC* IHCs with a fluid jet to deflect the hair bundle. Using a high-speed imaging system we were able to also measure the hair bundle deflection during the fluid-jet stimulation (Fig. 6a). We chose fluid-jet stimulation as opposed to stiff probe stimulation, since the fluid jet can stimulate all the stereocilia in IHC hair bundles without biasing stereocilia. These are problems the stiff probe has due to the bulky shape to stimulate IHCs and the necessity of touching the stereocilia. To determine if MYO7A-C is involved in set point regulation of the MET channel, we used sinusoidal hair bundle stimulation to quantify the MET channel resting open probability

($P_o$). To increase the sensitivity of our assay, we used an intracellular solution containing 10 mM BAPTA, a calcium chelator, which increases the WT resting open probability (Fig. 6b)[38,39]. The resting $P_o$ of the MET channel in IHCs was severely affected in *Myo7a-ΔC* mice, decreasing from $8.6 \pm 3.9\%$ in WT to $3.3 \pm 1.2\%$ in *Myo7a-ΔC* IHCs ($p = 5.7e-5$) (Fig. 6b, c, arrowheads indicate near absent negative current in IHC *Myo7a-ΔC*). In basal OHCs, in which MYO7A levels were unaffected in *Myo7a-ΔC* mice, we would not expect a change in resting open probability. Indeed, we found no change in resting $P_o$ (Fig. 6c, $p = 0.60$). Next, to examine changes to the current vs displacement (activation) curves, we measured MET currents in response to fluid-jet step stimuli (Fig. 6d). Peak currents showed no significant change between WT and mutant IHCs ($p = 0.43$) (Fig. 6f), but the activation curves between the WT and mutant IHCs showed a rightward shift ($x_o$, $p = 0.011$) and the resting $P_o$ was significantly reduced (Fig. 6d, e, g; $p = 6.0e-4$). No significant differences were observed with similar experiments in basal OHCs, which served as an additional control (Supplementary Fig. 3; $x_o$, $p = 0.39$ and resting $P_o$, $p = 0.67$). Of note, the resting $P_o$ of ~15% in WT OHCs (as opposed to 50% in some reports[38–40]) is due to the high extracellular calcium concentration and high intracellular calcium buffer used for the present recordings. We ensured that all measurements have clamp time constants that are twofold shorter than the process measured. These data are consistent with MYO7A-C having a role in setting tip-link tension.

Decreases in tip-link tension in OHCs have been connected with slowing of MET activation[24]. To determine whether MET channel activation is altered in IHCs of *Myo7a-ΔC* mice, we analyzed the step response kinetics. We determined the $t_{10\%}$, which is the time for the current and motion to reach 10% of their maximum value (in Fig. 6h–j). Across the entire stimulus range (50, 75, 100% $P_o$), *Myo7a-ΔC* IHCs exhibited a significant delay in the $t_{10\%}$ of the onset of the current (two-way analysis of variance (ANOVA), $p = 4.4e-6$) (Fig. 6j), but not the motion (two-way ANOVA, $p = 0.48$) (Fig. 6i). Since there is no delay in the motion, the hair bundle begins its movement at the same time. Even though the fluid-jet stimulus has a rise time of ~0.5 ms[41], it was fast enough to dissect differences in current onset between WT and *Myo7a-ΔC* hair cells. The delay in the current onset is consistent with the tip-link tension being slack prior to the onset of the stimulus requiring extra time to engage and provide tension on the MET channel.

The opposite effect occurred for the $t_{10–90}$, which is the time for the current and motion to go from 10 to 90% of their maximum value. Here, the rise of the current was comparable between WT and mutant IHCs, but the rise of motion was prolonged significantly (two-way ANOVA, current: $p = 0.24$, motion: $p = 4.0e-4$). The effect on the kinetics of the hair bundle motion suggests some mechanical changes in the hair bundle due to the loss of MYO7A-C. We previously determined that the fluid-jet force plateaus ~0.5 ms after stimulus onset, and the hair bundle will continue to move even after the force has plateaued[41]. The continued motion, which we term "creep," allows us to measure mechanical changes in the hair bundle. We fit the creep with a double-exponential decay function, which we previously found to provide a better fit than a single exponential decay[41]. Comparing WT and *Myo7a-ΔC* mice (Supplementary Fig. 4), we found that the creep was greater in *Myo7a-ΔC* mice with a larger contribution of the slower creep component (two-way ANOVA: $A_2$—magnitude of the slower creep component $p = 0.043$; ($A_2/(A_1 + A_2)$—relative magnitude of the slower creep component, $p = 2.2e-4$; ($A_1 + A_2)/y_0$—relative magnitude of the creep compared with the total displacement of the hair bundle, $p = 1.0e-4$). The differences

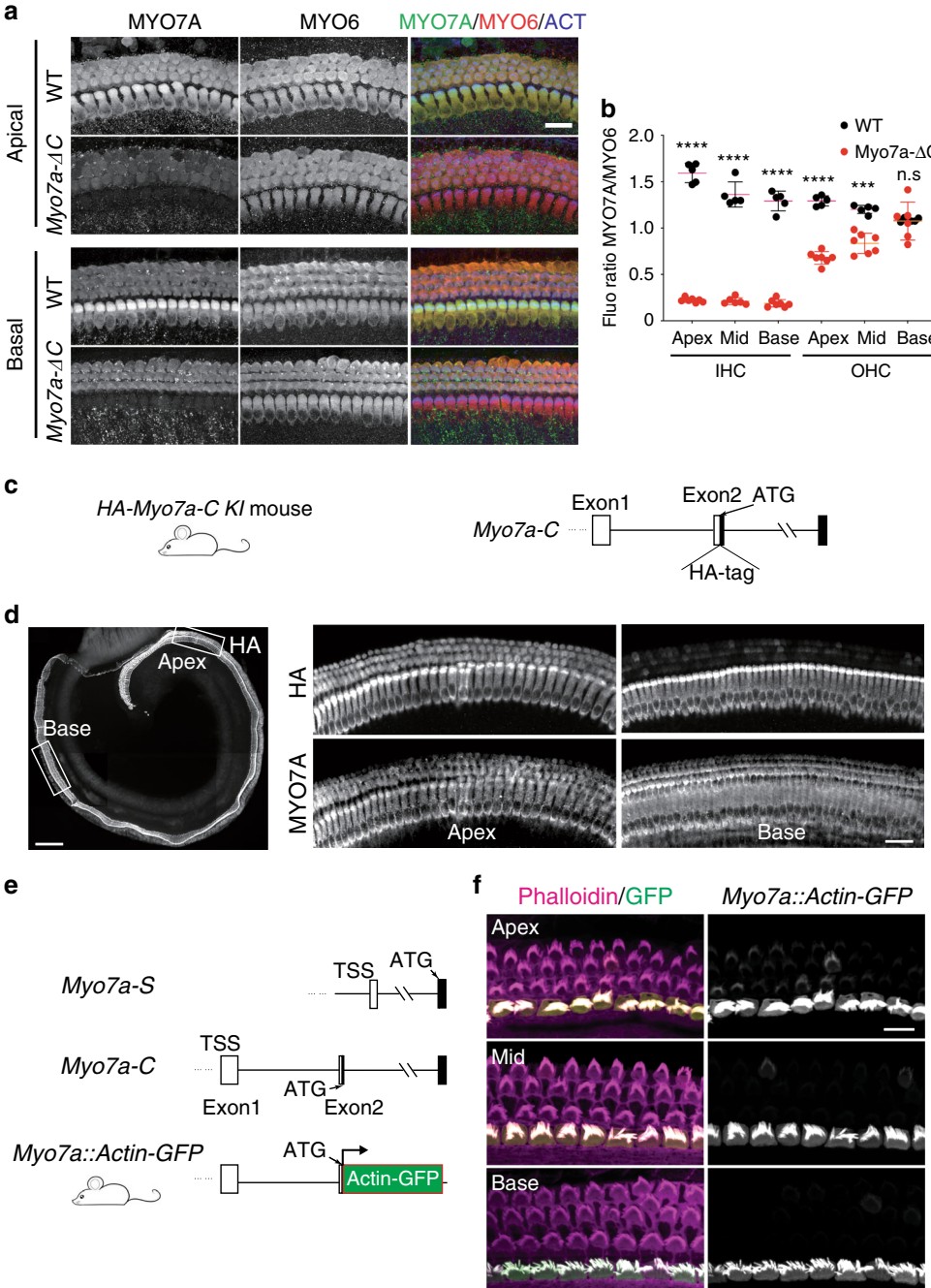

**Fig. 3 MYO7A-C is expressed primarily in IHCs and in a tonotopic gradient in OHCs, with decreasing expression toward the cochlear base. a** MYO7A and MYO6 immunoreactivity in apical and basal turns of organ of Corti of P5 *Myo7a-ΔC* and WT mice (scale bar: 20 μm). **b** MYO7A immunoreactivity was quantified in apical, middle, and basal turns by normalizing to MYO6 immunoreactivity. Compared with WT, MYO7A levels in *Myo7a-ΔC* mice were reduced by ~85% (*p* < 1e−4) in all IHCs, and by ~30% and 52% (*p* < 1e−3 and <1e−4) in middle and apical-turn OHCs. Each data point in **b** corresponds to a mean MYO7A/MYO6 immunofluorescence ratio (mean of 12 IHCs and 36 OHCs per organ and position). Error bars indicate SD. *N* (animals) = 5 in WT and 7 in *Myo7a-ΔC*. *p* values in **b** were derived from two-tailed, unpaired *t*-tests. **c** Design of the *HA-Myo7a-C KI* mouse. HA-tag was knocked-in after the ATG of *Myo7a-C* by CRISPR-mediated gene editing. **d** Organ of Corti of an *HA-Myo7a-C KI* mouse, stained with a HA-specific antibody. The white boxes indicate approximate positions of the apical and basal cochlear regions shown on the right. HA immunoreactivity, representing the expression of MYO7A-C, is strong in all IHCs. In OHCs, HA immunoreactivity is readily detected at the cochlear apex, but decreases toward the base (scale bars: 200 μm in whole organ of Corti image, 20 μm in magnified panels). **e** The genomic position from which the *Myo7a* promoter used in the *Myo7a::Actin-GFP* transgenic mouse is derived. **f** Actin-GFP signal and phalloidin reactivity in the organ of Corti of *Myo7a::Actin-GFP* transgenic mice at P6. The Actin-GFP signal was predominantly observed in the IHCs. Actin-GFP signal was detected at low levels in the apical OHCs, and decreased tonotopically toward the basal end of the cochlea (scale bar: 10 μm). Source data are provided in the Source Data file.

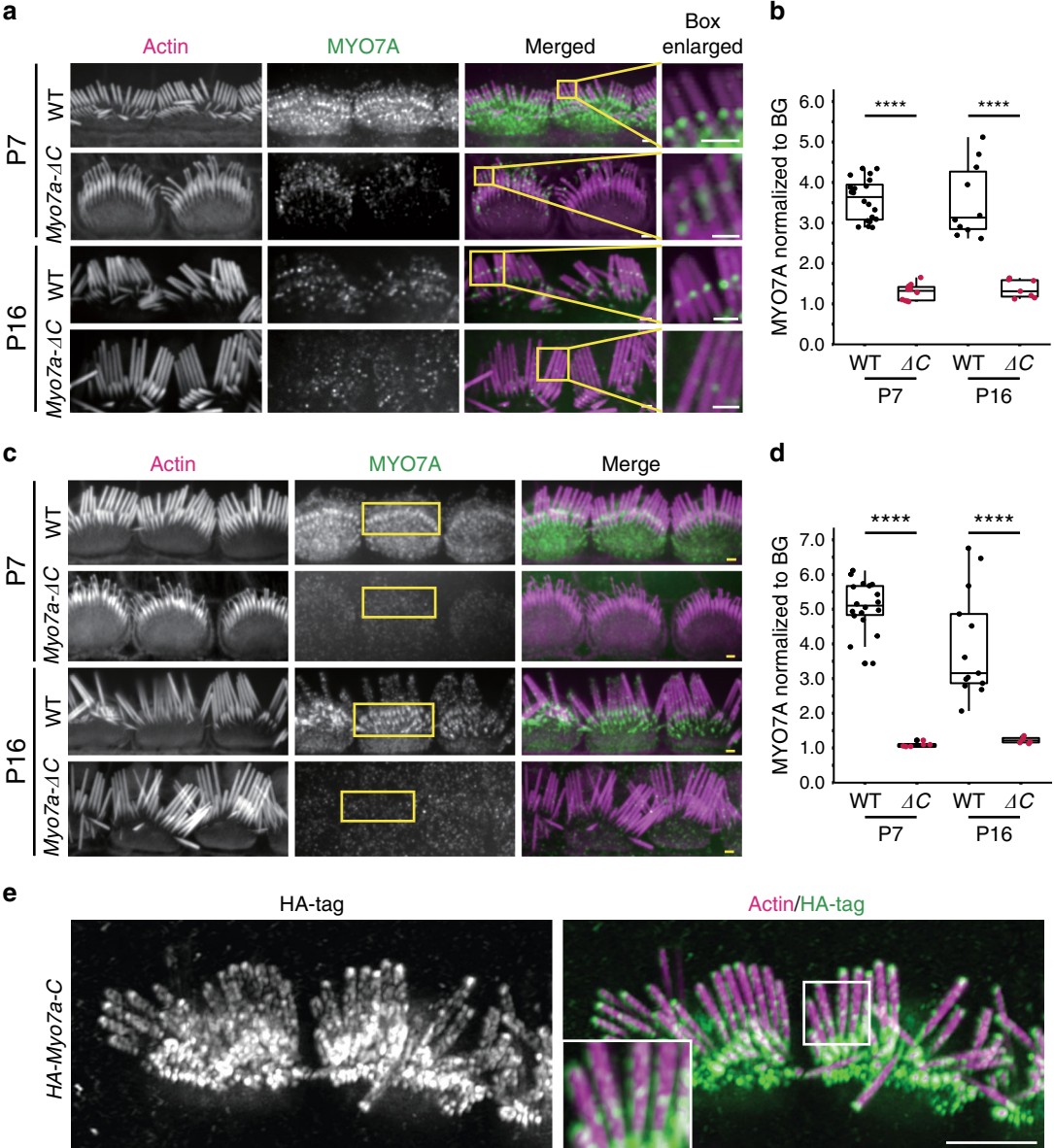

**Fig. 4 Reduced levels of MYO7A at the UTLD and stereocilia base of *Myo7a-ΔC* mice. a** MYO7A immunoreactivity at the UTLD in WT and *Myo7a-ΔC* mice at P7 and P16. MYO7A is detected at the UTLD of *Myo7a-ΔC* IHCs, but the intensity is reduced compared with WT (scale bars: 1 μm). **b** MYO7A in the UTLD of *Myo7a-ΔC* mice decreased by 68% at P7 ($p < 1e-3$) and 58% at P16 ($p < 1e-3$) compared with WT. The fluorescence intensity of MYO7A puncta was normalized against a dark background near the UTLD staining in the same image and z-section. Mean normalized intensities of MYO7A at the UTLD, averaged per cell: P7: WT = 3.58 ± 0.51, *Myo7a-ΔC* = 1.28 ± 0.21, $p = 2.73e-16$; P16: WT = 3.59 ± 1.00, *Myo7a-ΔC* = 1.37 ± 0.23, $p = 4.30e-05$. *N* = number of cells. Numbers analyzed per genotype and age (UTLDs; cells; animals): WT, P7 (157; 20; 4), P16 (73; 10; 4). *Myo7a-ΔC*, P7 (62; 10; 4), P16 (35; 7; 4). **c** MYO7A immunoreactivity at the stereocilia base of IHCs of WT and *Myo7a-ΔC* at P7 and P16 (scale bars: 1 μm). **d** Quantification of MYO7A at the stereocilia base. MYO7A puncta were normalized against a dark background in the image. Mean normalized intensities of MYO7A at the base, averaged per cell: P7: WT = 5.09 ± 0.71, *Myo7a-ΔC* = 1.093 ± 0.068, $p = 6.98e-15$; P16: WT = 3.96 ± 1.54, *Myo7a-ΔC* = 1.23 ± 0.089, $p = 3.35e-05$. *N* = number of cells. Numbers analyzed per genotype and age (stereocilia bases; cells; animals): WT P7 (406; 18; 4), P16 (187; 13; 4); *Myo7a-ΔC*: P7 (64; 6; 4), P16 (79; 6; 4). *p* values in **b** and **d** were derived from two-tailed, unpaired *t*-tests. Boxplots show medians, 25th and 75th percentiles as box limits, and minima and maxima as whiskers. **e** HA-immunofluorescence resolved the membrane-adjacent localization of HA-MYO7A-C in the *HA-Myo7a-C KI* mice, as well as MYO7A enrichment at the predicted site of the UTLD and the stereocilia base (and tips in some cases). The inset shows the HA-tag signal at the UTLD at higher magnification (scale bar: 5 μm). Source data are provided in the Source Data file.

in motion suggest that the mechanical properties of the hair bundle are altered in *Myo7a-ΔC* mice.

All taken together, the characteristics of both the MET current and hair bundle motion in *Myo7a-ΔC* IHCs are consistent with the hypothesis that MYO7A-C generates tension within the IHC MET complex.

***Myo7a-ΔC* mice develop rapidly progressing hearing loss**. We next examined the hearing performance in the *Myo7a-ΔC* mice at various ages. Auditory brainstem response (ABR) thresholds at early ages (P17) were only mildly affected (ABR thresholds were shifted between 10–30 dB depending on test frequency), but thresholds increased with age, leading to near profound deafness

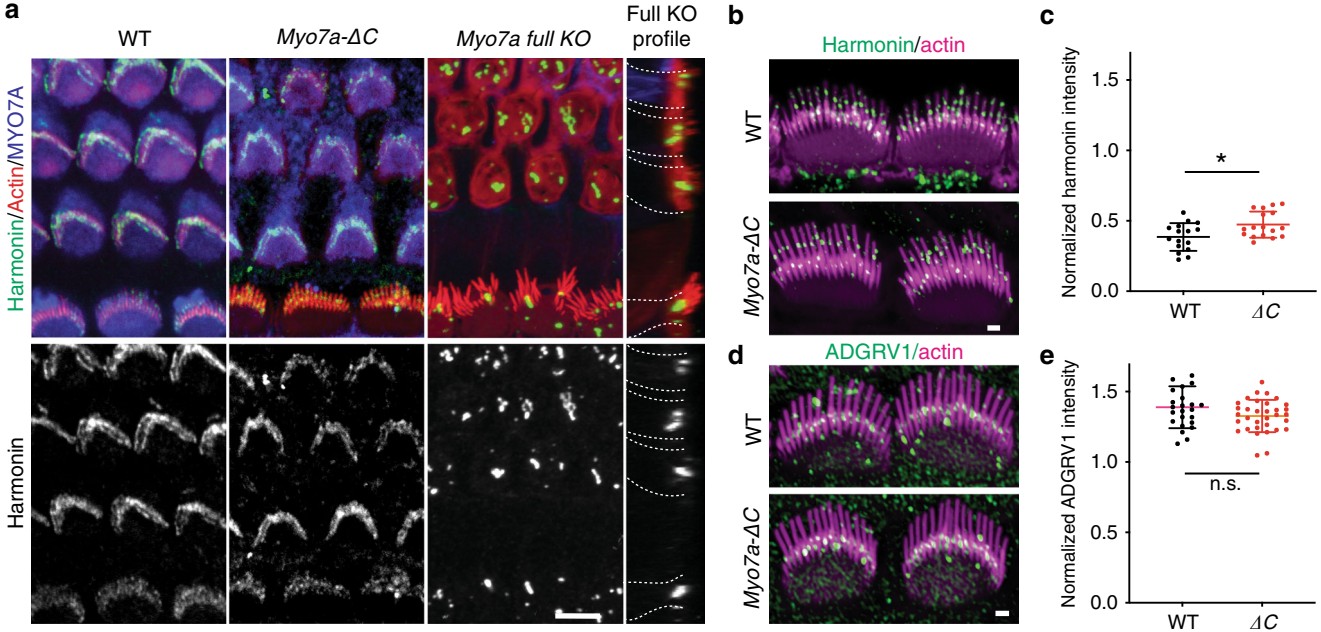

**Fig. 5 Harmonin and ADGRV1 localization at the UTLD and the stereocilia base are not decreased in *Myo7a-ΔC* mice. a** WT and *Myo7a-ΔC* organ of Corti immunostained for harmonin, F-actin, and MYO7A. Harmonin staining was indistinguishable between WT and *Myo7a-ΔC*, while severe mislocalization (formation of aggregates in the cuticular plate) was observed in *Myo7a full KO* mice (scale bar: 10 μm). **b, c** Harmonin immunoreactivity at the UTLDs in WT and *Myo7a-ΔC* IHCs at P7. Compared with WT, harmonin signal is slightly increased at the UTLD of *Myo7a-ΔC* IHCs. For quantification, harmonin fluorescence intensities in stereocilia (UTLDs of longest stereocilia only) were measured using ImageJ, normalized to phalloidin reactivity at the same site, and a mean intensity determined for each cell. Normalized harmonin intensities (mean ± SD): WT = 0.40 ± 0.092, *Myo7a-ΔC* = 0.47 ± 0.093, $p = 2.34e{-}2$ (two-tailed, unpaired *t*-tests). $N$ = number of cells. Analyzed numbers (UTLDs; cells; animals): WT (167; 16; 3), *Myo7a-ΔC* (209; 17; 4) (scale bar: 1 μm). **d, e** ADGRV1 immunoreactivity at the stereocilia base was not significantly changed in *Myo7a-ΔC* IHCs at P7 compared with WT. ADGRV1 immunofluorescence intensity (from middle cochlear region) was determined for each cell and normalized to background immunofluorescence in the cuticular plate region of the same cell. Normalized ADGRV1 intensities (mean ± SD): WT = 1.39 ± 0.49, *Myo7a-ΔC* = 1.32 ± 0.16, $p = 0.08$ (two-tailed, unpaired *t*-tests). $N$ = number of cells. Analyzed numbers (cells; animals): WT (24; 7), *Myo7a-ΔC* (33; 7) (scale bar: 1 μm). Data are plotted as mean ± SD. Source data are provided in the Source Data file.

at the age of 9 weeks (Fig. 7a). In contrast, distortion product otoacoustic emissions (DPOAEs), a measure of OHC function, were unaffected at all ages tested (Fig. 7b). Despite a significant reduction of MYO7A in vestibular utricles (by 63%) (Fig. 2b), the *Myo7a-ΔC* mice exhibited no obvious circling behavior at four weeks of age, which is in contrast to the *full Myo7a KO* mice that exhibited extensive circling and head-bobbing behavior.

**Transducing rows of stereocilia degenerate in *Myo7a-ΔC* IHCs.** *Myo7a-ΔC* mice presented with a hearing loss phenotype that was consistent with the specific reduction of MYO7A in IHCs and not OHCs. We therefore used SEM analyses to investigate ultrastructural correlates of IHC dysfunction in mature mice. Previous studies have shown that loss of MET causes stereocilia regression[28,42]. We therefore predicted similar stereocilia phenotypes in the *Myo7a-ΔC* IHCs. At 3 weeks of age, when the ABR thresholds in the *Myo7a-ΔC* mice is shifted by 10–30 dB compared with WT, the stereocilia of IHCs appear similar to the WT controls, but closer analysis revealed that the numbers of the shortest stereocilia (in the third row) were significantly reduced (Fig. 8a, b). This phenotype progressed with age, and at 8 weeks, most third-row stereocilia were absent in the *Myo7a-ΔC* mice (Fig. 8a, b). No differences were detected in the numbers of the first and second row of IHC stereocilia (Fig. 8a, b).

We next measured the lengths of the longest and second row of stereocilia. Interestingly, both rows, as quantified at 6 weeks, were slightly longer than WT counterparts (Fig. 8c, d). Importantly, the second row of stereocilia exhibited a marked loss of prolate tips at 6 and 8 weeks of age (Fig. 8e, f), consistent with a reduction

in tip-link tension[43]. At no time tested were OHC morphologies affected in *Myo7a-ΔC* mice (Fig. 8g, h). In summary, the morphological phenotype observed in the mechanosensory hair bundles of *Myo7a-ΔC* mice is consistent with a reduced function of the tip-link tensioning mechanism specifically in IHCs.

## Discussion

This is the first study to report that multiple isoforms of MYO7A are expressed in the cochlea. The isoforms display distinct hair cell-type expression patterns that, in OHCs, vary tonotopically. Analyses of KO and KI mouse models revealed that the canonical isoform MYO7A-C is predominantly expressed in IHCs, while in OHCs, MYO7A-C is expressed in a tonotopic manner, its levels decreasing from the apex to the base of the cochlea. Genomic databases indicate the existence of an alternative isoform we tentatively named MYO7A-S. Since its coding sequence overlaps with the canonical MYO7A-C isoform and cannot be specifically tagged or deleted, direct evidence for this isoform is lacking. Assuming that the MYO7A expression remaining in the *Myo7a-ΔC* mice represents MYO7A-S, its expression is expected to be inversely correlated to that of MYO7A-C.

The proposed expression pattern of MYO7A-C mirrors the activity of a previously characterized *Myo7a* promoter, which was derived from a genomic region upstream of the translational start site of the canonical isoform[29,30]. In the initial study[44], this *Myo7a* promoter drove strong expression in IHCs, but weak expression in OHCs. Variants of this canonical *Myo7a* promoter are used to drive transgene expression in hair cells, in both transgenic mice and viral constructs. Virally transduced OHCs

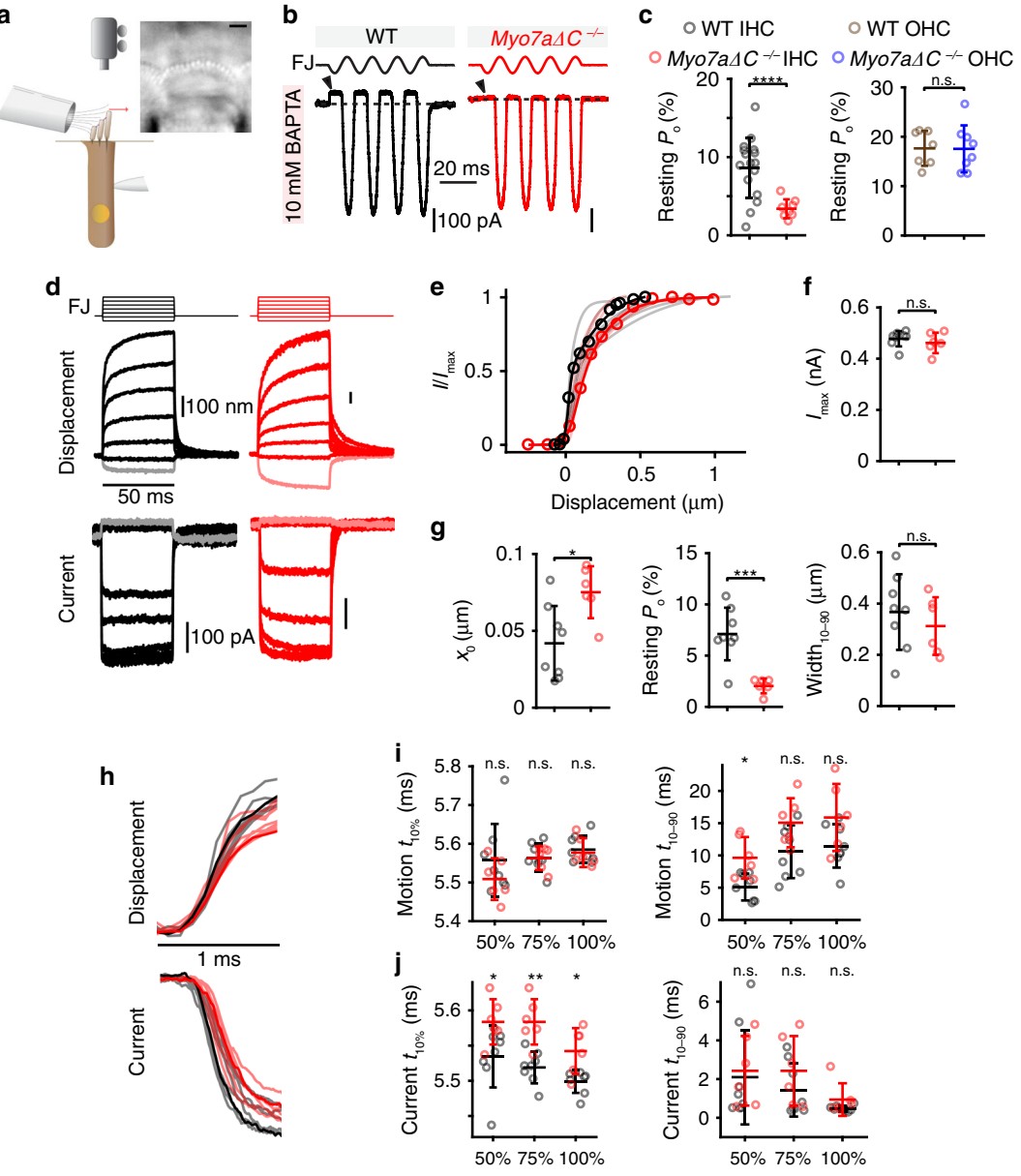

**Fig. 6 Reduced MET resting $P_o$ and slowed MET currents in *Myo7a-ΔC* IHCs. a** Illustration of setup with fluid-jet stimulation, patch-clamp electrophysiology, and high-speed imaging with an IHC image (scale bar: 2 μm). **b** MET responses of IHCs to sinusoidal bundle stimulation. Current shut off during the negative phase of the stimulation (arrowheads) represents the resting current. Resting current divided by maximum MET current estimates resting open probability. **c** Resting $P_o$ decreased ($p = 7.5e{-}5$, number of cells (animals); WT: $n = 16$ (11); *Myo7a-ΔC*: $n = 8$ (7)) in *Myo7a-ΔC* IHCs. Similar experiments in cochlear basal turn OHCs showed no change in resting $P_o$ ($p = 0.88$, two-tailed, unpaired *t*-test, WT: $n = 12$ (7); *Myo7a-ΔC*: $n = 14$ (8)). **d** MET responses of IHCs to step-like force stimulation are shown with the fluid-jet driving voltage waveform (FJ). Largest negative stimuli are shaded lighter. **e** Current vs displacement plots (activation curves) for the data from **d** shows the *Myo7a-ΔC* cell is right-shifted. Light-colored traces show curve fits from all cells. **f** Peak currents showed no change ($p = 0.43$). **g** $x_o$ in *Myo7a-ΔC* mice are right-shifted as compared with controls ($p = 0.011$), consistent with reduced resting $P_o$ ($p = 6e{-}4$, two-tailed, unpaired *t*-test). **h** Regarding the onset, no difference in the displacement rise but a delay in the current onset was found. Steps eliciting ~75% peak current are shown. **i** For the steps eliciting ~50, ~75, and ~100% peak current, no difference in the motion onset time (time to reach 10% maximum motion, $t_{10\%}$, $p = 0.48$, two-way ANOVA) but a difference in rise time (time from 10 to 90% maximum motion, $t_{10-90}$) was found ($p = 4.0e{-}3$, two-way ANOVA). **j** Conversely, current onset time was delayed in *Myo7a-ΔC* mice by ~0.05 ms ($p = 4.4e{-}6$, two-way ANOVA), with no change in current rise time ($t_{10-90}$, $p = 0.24$, two-way ANOVA). Summary plots in **c**, **f**, **g**, **i**, and **j** are represented as mean ± SD. WT: $n = 8$ (7), *Myo7a-ΔC*: $n = 6$ (6) for panels **f**, **g**, **i**, and **j**. Data are in the Source Data file.

exhibited inconsistent transgene expression and functional rescue[45,46]. Our findings of multiple isoforms and their promoters now explain these past inconsistencies.

In *Myo7a-ΔC* IHCs, the expression of MYO7A happens to be at a critical level, high enough to enable normal hair bundle development, but sufficiently low to expose a hypomorphic phenotype in hair cell MET. The MET current phenotype in *Myo7a-ΔC* IHCs

is consistent with a model in which MYO7A operates as the motor that tensions the tip link. Hypofunction of MYO7A causes a slackening of the tip link, leading to a reduction of $P_o$. Upon stimulus onset, the initial bundle motion is used to make up for the slack in the tip link, causing a delayed onset of the MET current.

*Myo7a-ΔC* mice develop a progressive form of hearing loss. Based on the mild MET current phenotype in P7 *Myo7a-ΔC*

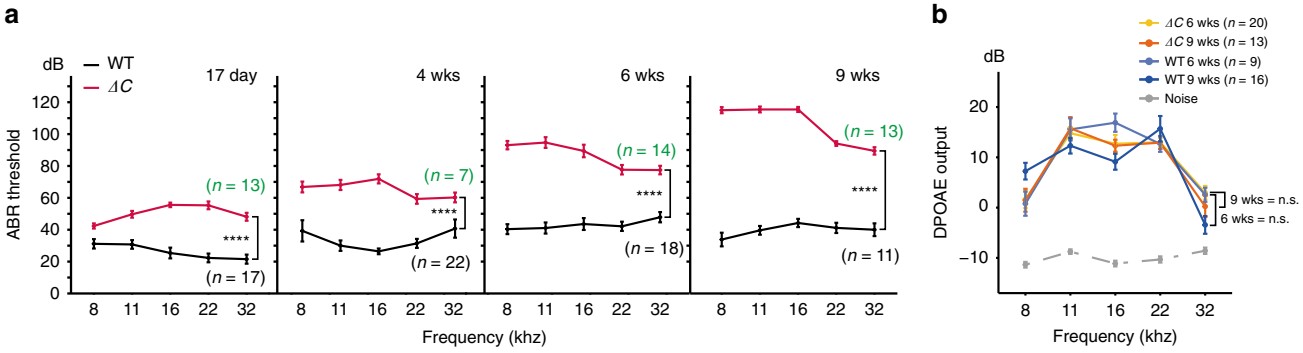

**Fig. 7 Progressive hearing loss in *Myo7a-ΔC* mice. a** ABR thresholds in *Myo7a-ΔC* and age-matched WT controls. Threshold differences increase progressively with age. p values for 17 days = 3.38e−29, 4 weeks = 1.62e−21, 6 weeks = 5.68e−44, and 9 weeks = 2.22e−54 (two-way ANOVA between *Myo7a-ΔC* and age-matched WTs). **b** No significant difference in DPOAE output between WT and *Myo7a-ΔC*. p values for 4 weeks = 0.69 and for 9 weeks = 0.83 (two-way ANOVA between *Myo7a-ΔC* and WT). Error bars indicate standard error of the mean and n = number of animals.

IHCs, it is not surprising that hearing thresholds are not severely affected in young animals. With increasing age, MYO7A hypofunction might cause significant MET dysfunction, which might affect hair bundle morphology and hearing. One must be cautious however to interpret the adult hearing loss phenotype solely from the angle of hair cell MET. MYO7A is widely localized in the hair cell, suggesting involvement in a range of yet unknown cellular processes. It is therefore possible that MYO7A hypofunction causes cellular dysfunction that are MET-independent, and that those deficits contribute to the progressive hearing loss.

In our study, reduced resting $P_o$ serves to indicate a less active tip-link motor. It is important to discuss the validity and robustness of this connection. In the *Myo7a-ΔC* IHCs, MYO7A is not only reduced at the UTLD, but also at the stereocilia base. Although we did not detect significant changes in the level or localization of the ankle-link component ADGRV1[31], it is possible that the stiffness of the stereocilia base, thus pivot stiffness, is affected in the *Myo7a-ΔC* IHCs. In fact, analysis of stereocilia motion in response to fluid-jet stimuli suggest that the mechanical properties, and potentially stiffness of the hair bundle, are altered in *Myo7a-ΔC* mice. The crucial question is whether changes in such additional "springs" in the system might influence tip-link tension and resting $P_o$ indirectly. A judgment on this depends on the interpretation of the existing hair cell MET model. According to a generally accepted assumption[47], the tip link is tensed by a motor protein that pulls with a certain force (e.g., the term $F_m$ in refs. [48–51]). This force is maintained (force-clamped) regardless of the stiffness of other springs in the system. Mechanistically, this might be achieved by the motor climbing up the actin ladder until its stall force is reached. Experimentally, this interpretation appears to be indirectly supported by the phenotype of the *Triobp KO* mice, in which, even though the pivot stiffness is severely reduced, hair cell MET currents showed no changes in the activation curve, suggestive of unchanged resting $P_o$[52]. Based on this reasoning, we suggest that resting $P_o$, regardless of changes in overall hair bundle stiffness, may serve as a reasonably robust proxy for tip-link motor activity. The hair bundle however is a complex mechanical system, and present assumptions about the tip-link motor might be simplistic. For example, the tip-link motor might not be able to adjust to severe reduction of the bundle stiffness if its upward mobility is limited, as was previously suggested[21]. Direct mechanical measurements of tip-link tension, using previously described methods[53,54], will provide more conclusive data on whether the reduction in resting $P_o$ in *Myo7a-ΔC* IHCs is indeed accompanied by reduced tip-link tension.

One outstanding question in the hair cell field concerns the molecular correlate of slow adaptation. Several studies have advanced a model in which myosin motors, slipping down the F-actin core, might provide the mechanism for slow adaptation, adjusting tip-link tension to ensure an optimal sensitivity range[16–20]. While not addressed in our present study, the *Myo7a-ΔC* mice provide a suitable model to address this question for IHCs.

Two myosins, MYO1C and MYO7A, have been proposed to function as the tip-link motor. Evidence for MYO1C only applies to vestibular hair cells so far and does not prove conclusively that MYO1C directly tensions the tip link[17,19]. Recent localization evidence places MYO7A at the correct position, the UTLD[21]. Providing functional evidence implicating MYO7A in controlling tip-link tension has been more challenging, however, because complete loss-of-function impairs stereocilia development[55]. Experiments using a *Myo7a* KO mouse model were likely misinterpreted due to the unexpected presence of an additional mechanically gated current[20,56,57]. Our present study is consistent with a more recent study that used a conditional knockout approach to study MYO7A's role in MET in intact hair cells[58]. In *Myo7a*fl/fl*Myo15-Cre*+/− mice, the *Myo15* promoter drives Cre-mediated *Myo7a* gene deletion in hair cells starting around P4, allowing hair cells to develop normally. MYO7A levels presumably reached critically low levels after P14 in IHCs, at which point electrophysiological recordings revealed a reduction in resting $P_o$, consistent with our results. Due to slow protein turnover after conditional deletion, functional experiments on MYO7A-depleted hair cells were limited to mice aged P14 and older. In comparison, our *Myo7a-ΔC* mice enable experiments at early postnatal ages and importantly, allow the investigation of the functional significance of MYO7A isoforms. The interpretation of the *Myo7a-ΔC* mouse phenotype needs to consider the fact that this mouse line is not a mere MYO7A hypomorph, but null for a specific isoform. It is therefore possible that some aspects of the described phenotype are caused by a combination of a depletion of overall MYO7A levels and the loss of isoform-specific functions. In summary, the conditional and isoform-specific KO mouse models are complementary tools in studying MYO7A's role in hair cell function.

With respect to the effect of MYO7A depletion on hair bundle morphology, it is noteworthy that *Myo7a-ΔC* IHCs display normal hair bundle development. We can therefore conclude that, at least during development, the remaining levels of MYO7A can compensate for the loss of MYO7A-C. Mature *Myo7a-ΔC* mice, however, develop a highly specific set of hair bundle defects in IHCs. Interestingly, the characteristics of those defects differ in the two transducing stereocilia rows: in the third row, stereocilia degeneration is evident, possibly consistent with previous studies showing stereocilia regression after loss of MET[28,43]. In contrast,

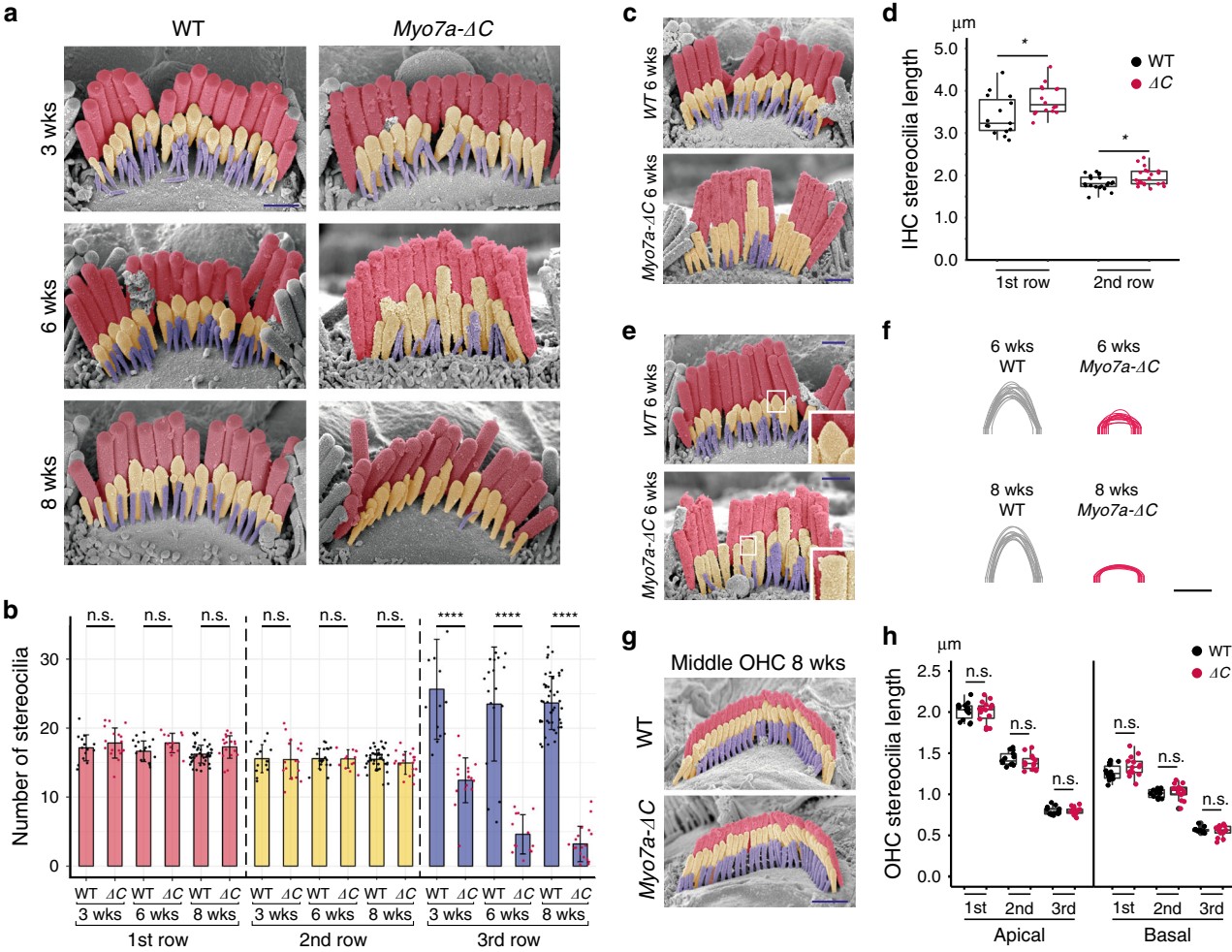

**Fig. 8 IHC stereocilia degenerate in *Myo7a-ΔC* mice. a, b** SEM analysis at 3, 6, and 8 weeks. The first, second, and third row of stereocilia are shown in red, yellow, and blue, respectively. Number of third row stereocilia in *Myo7a-ΔC* IHCs decreases progressively: mean third row stereocilia number ± SD per cell (number of cells; animals): 3 weeks WT = 25.64 ± 7.25 (14;3), *Myo7a-ΔC* = 12.44 ± 3.26 (18;3), p = 7.28e−23. 6 weeks WT = 22.65 ± 2.79 (36;6), *Myo7a-ΔC* = 4.50 ± 3.03 (20;5), p = 7.28e−23. 8 weeks WT = 20.83 ± 4.54 (35;3), *Myo7a-ΔC* = 3.84 ± 2.54 (31;3), p = 9.41e−26. **c, d** First and second row stereocilia of *Myo7a-ΔC* IHCs are longer than WT counterparts. SEM images are collected from the middle turn. Mean stereocilia lengths at 6 weeks (averaged by cell) ± SD. First row: WT = 3.41 ± 0.47, *Myo7a-ΔC* = 3.78 ± 0.35, p = 0.021; analyzed numbers (stereocilia; cells; animals) for WT (95;15;5) and *Myo7a-ΔC* (86;16;5). Second row: WT = 1.83 ± 0.17, *Myo7a-ΔC* = 1.96 ± 0.21, p = 0.033. Analyzed numbers (stereocilia; cells; animals) for WT (116;20;3) and *Myo7a-ΔC* (159;21;3). **e, f** Loss of prolate tips in second row stereocilia in *Myo7a-ΔC* IHC. Stereocilia tips were traced and superimposed. **g, h** OHC bundles at 8 weeks were not affected in *Myo7a-ΔC* mice. SEM-based quantification of stereocilia lengths: basal OHCs: First row: WT = 1.27 ± 0.094, *Myo7a-ΔC* = 1.34 ± 0.12, p = 0.089. Second row: WT = 1.01 ± 0.050, *Myo7a-ΔC* = 1.032 ± 0.096, p = 0.52. Third row: WT = 0.57 ± 0.031, *Myo7a-ΔC* = 0.55 ± 0.069, p = 0.58. Analyzed numbers (stereocilia; cells; animals) for WT (188;14;5) and for *Myo7a-ΔC* (153;12;4). Apical OHCs: First row: WT = 2.01 ± 0.10, *Myo7a-ΔC* = 1.99 ± 0.13, p = 0.688. Analyzed numbers (stereocilia; cells; animals) for WT (131;13;3) and for *Myo7a-ΔC* (137;18;4). Second row: WT = 1.43 ± 0.082, *Myo7a-ΔC* = 1.39 ± 0.082, p = 0.21. Analyzed numbers (stereocilia; cells; animals) for WT (99;13;4) and for *Myo7a-ΔC* (94;14;3). Third row: WT=0.80 ± 0.049, *Myo7a-ΔC* = 0.80 ± 0.045, p = 0.95. Analyzed numbers (stereocilia; cells; animals) for WT (99;13;4) and for *Myo7a-ΔC* (94;14;3). Boxplots show medians, 25th and 75th percentiles as box limits, and minima and maxima as whiskers. p values in **b**, **d**, and **h** were derived from two-tailed, unpaired t-tests (error bars indicate SD and n = number of cells). All scale bars: 1 μm (except in **e** and **f**: 0.2 μm). Source data are provided in the Source Data file.

the second row presents with a more complex phenotype, a reduction in prolate stereocilia tips and slightly longer stereocilia. The former is consistent with a reduction in resting tension[43], while the latter phenotype, although somewhat paradoxical compared with the phenotype of the third row, is consistent with previous studies reporting that stereocilia in *Myo7a*[4626SB] mice are longer[29,59].

Possibly the most important question arising from our study is why multiple isoforms of MYO7A are expressed in cochlear hair cells. Assuming MYO7A is indeed the tip-link tensioning motor, we propose that the differential expression of MYO7A may contribute to the shaping of MET properties, by modulating tip-

link tension and resting $P_o$. Several conclusions from previous studies are consistent with our hypothesis: first, it is well established that IHCs and OHCs have distinct MET current properties. Of most relevance for our study, MET in OHCs operates with a resting $P_o$ of ~50%, while the resting $P_o$ in IHCs is ~10–20%[38–40]. The tectorial membrane is unlikely to play a crucial role in this, since the differences persist in recordings in which the tectorial membrane is removed. Differing expression levels of intracellular calcium buffers[39,60] and MET channel components[42] were proposed as potential mechanisms underlying these differences, but direct evidence was not provided. Second, a recent study in rats reported that tension in individual tip links is generally lower in

IHCs compared with OHCs[54]. Furthermore, tip-link tension in OHCs was found to increase sharply along the tonotopic axis[54]. Taken together, it is tempting to hypothesize a model in which the differential expression of the two MYO7A isoforms, each endowed with distinct motor properties, contributes to the differences in tip-link tension and resting $P_o$ in IHCs and OHCs, and along the tonotopic axis in OHCs.

The kinetics of MYO7A motors from various organisms have been studied, demonstrating that MYO7A is a slow-processive motor with a high duty ratio[22,61–63]. These are characteristics consistent with a role of MYO7A in mediating tonic tensioning of the tip-link complex. Whether the distinct N-termini of the MYO7A isoforms indeed differentially affect motor properties will have to be addressed in in vitro experiments using purified proteins, but there is precedent in other myosins: the isoforms of MYO1C, also distinguished by short N-terminal extensions, were shown to exhibit differences in motor activity[64]. A similar N-terminal extension also affects mechanical tuning in MYO1B[65,66].

Lastly, the significance of the two MYO7A isoforms for human deafness remains to be shown. The 11-aa peptide specific for MYO7A-C is present and sequence conserved in humans, but its short size makes a deafness mutation specifically affecting the long MYO7A isoform unlikely. However, mutations in the UTR and *cis*-regulatory domains of the different *MYO7A* isoforms might affect their expression, which would be predicted to affect human hearing.

## Methods

**Animal care and handling.** The care and use of animals for all experiments described conformed to NIH guidelines. Experimental mice were housed with a 12:12 h light:dark cycle with free access to chow and water, in standard laboratory cages located in a temperature and humidity-controlled vivarium. The protocols for care and use of animals was approved by the Institutional Animal Care and Use Committees at the University of Virginia, the University of Colorado Denver and at the National Institute on Deafness and Other Communication Disorders. All above-mentioned institutions accredited by the American Association for the Accreditation of Laboratory Animal Care. C57BL/6J (Bl6, from Jackson Laboratory, ME, USA) mice and sibling mice served as control mice for this study. Neonatal mouse pups [postnatal day 0 (P0)–P5] were sacrificed by rapid decapitation, and mature mice were euthanized by $CO_2$ asphyxiation followed by cervical dislocation. The *Myo7a::beta-actin:GFP* mouse line expressing beta-actin-GFP from a Myo7a BAC transgene[29,30] was a gift from Dr Haydn Prosser (Wellcome Trust Sanger Institute, UK).

**Generation of Myo7a-ΔC, Myo7a full KO, and HA-Myo7a-C mice.** For CRISPR/Cas mediated generation of the mouse models, we used the online tool CRISPOR (http://crispor.tefor.net/crispor.py)[67] to select suitable target sequences to ablate the *Myo7a-C* isoform (AAGCATGGTTATTCTGCAGA, in exon 2). The same target was used for generating the *HA-Myo7a-C* mouse strain. Sequence of the repair template: GCCTGGGCTCAGGGCGTGCCATGGTCTCTTCCCACAGA GCTGTGTCTGGTCACTCCGGCAGGTGTGCTGACGTAGAAGCATG**TAC CCATACGATGTTCCAGATTACGCT**GTTATTCTGCAAAAGGTGAGTGCGT CTCCTCTCTCTCAGAGCTGCAGAGGGCCATGCTGGGTACCTCACATC CCACCCTGCA (HA-coding sequence in bold, flanked by mouse *Myo7a* locus-specific homology arms). For generating the *Myo7a full KO* mouse, the target CCTCCTCGTCTTCATCAGGC in exon 24 was used. To generate the corresponding single-guide (sg)RNAs, a PCR product from overlapping oligonucleotides (as described in the CRISPR online tool) was generated by T7 in vitro transcription. Oligonucleotide sequences: for *Myo7a-ΔC*: forward primer: GAAATTAATACGA CTCACTATAGAAGCATGGTTATTCTGCAGAGT TTTAGAGCTAGAAATA GCAAG and universal reverse primer: AAAAGCACCGAC TCGGTGCCACTT TTTCAAGTTGATAACGGACTAGCCTTATTTTAACTTGCTATTTCTAG CTCTAAAAC, and for *Myo7a full KO*: forward primer: GAAATTAATACGACT CACTATAGCCTCCTCGTCTTCATCAGGCGTTTTAGAGCTAGAAATA GCAAG and universal reverse primer. In vitro transcription was performed using the MAXIscript T7 kit (Life Technologies) and RNA was purified using the MEGAclear kit (Thermo Fisher Scientific, Waltham, MA). For production of genetically engineered mice, fertilized eggs were coinjected with Cas9 protein (PNA Bio, 50 ng/μl) and the sgRNA (30 ng/μl). Two-cell stage embryos were implanted on the following day into the oviducts of pseudopregnant ICR female mice (Envigo). Genotyping was performed by PCR amplification of the region of interest (*Myo7a-ΔC* forward primer: CTGAAGACTAAGTAGGAGTTTG; *Myo7a-ΔC* reverse primer: TAGACTGAGCTTTAATCAGAAG, *Myo7a full KO* forward primer: CCAGCCTAACGGTT AAGACA; *Myo7a full KO* reverse primer:

AGCTGGTCACCCTCATCGT), followed by Sanger sequencing. Founder mice were selected. The *Myo7a-ΔC* mouse harbors a reading frame-shifting 1-bp insertion in exon 2. The *Myo7a full KO* founder harbors a 26-nucleotide deletion in exon 24 (deleted sequence: CCTGCCTGATGAAGACGAGGAGGACC). *Myo7a-ΔC* mice were genotyped by Sanger sequencing, and *Myo7a full KO* mice by gelelectrophoretic size analysis (WT band 244 bps, KO band 218 bps).

**Immunofluorescence.** Inner ear organs were fixed in 3% paraformaldehyde (PFA, Electron Microscopy Sciences, PA) immediately after dissection for 20 min. Samples were washed three times with phosphate-buffered saline (GIBCO® PBS, Thermo Fisher Scientific, Waltham, MA) for 5 min each. After blocking for 2 h with blocking buffer (1% bovine serum albumin, 3% normal donkey serum, and 0.2% saponin in PBS), tissues were incubated in blocking buffer containing primary antibody at 4 °C overnight. The following antibodies were used in this study: rabbit polyclonal Myosin-VIIa antibody (catalog#: 25-6790, Proteus Biosciences Inc, Ramona, CA. 1:100), mouse monoclonal Myosin-VIIa antibody (Developmental Studies Hybridoma Bank, MYO7A 138-1, concentrate, 1:100), mouse monoclonal Myosin-VI antibody (A-9, Santa Cruz, 1:100), rabbit anti-Harmonin antibody (H3, obtained from Ulrich Mueller's lab)[24], rabbit anti-ADGRV1 antibody (obtained from Dominic Cosgrove's lab)[37], rabbit polyclonal Myosin-VIIa antibody used in Fig. 2 (PB206) was custom-generated and affinity purified against the immunizing MYO7A peptide LPGQEGQAPSGFEDLERGR[21], and rabbit anti-HA antibody (C29F4, Cell Signaling Technologies, Danvers, MA). Fluorescence imaging was performed using a Zeiss LSM880 or a Nikon microscope equipped with a 100×, 1.45NA objective and an Yokogawa confocal spinning disk attachment. Images were acquired with an Andor EMCCD camera using Nikon Elements software.

**Scanning electron microscopy.** Adult mice were euthanized by $CO_2$ asphyxiation before intracardiac perfusion with 2.5% glutaraldehyde (Electron Microscopy Sciences, Hatfield, PA) and 2% PFA. The otic capsule was dissected and incubated in postfixation buffer at 4 °C overnight (2.5% glutaraldehyde, in 0.1 M cacodylate buffer, with 3 mM $CaCl_2$). For neonatal mouse pups, the samples were dissected and treated with postfixation buffer immediately. The otic capsules from adult mice were incubated for 2 weeks in 4.13% EDTA for decalcification and then further dissected to expose the organ of Corti. Samples underwent the OTOTO procedure[68] and dehydrated by using gradient ethanol and critical point drying. After sputter coating with platinum, the samples were imaged on Zeiss Sigma VP HD field emission SEM using the in-lens secondary electron detector.

**Determination of stereocilia length from SEM stereo images.** Previous studies have reported methods to calculate stereocilia length from SEM stereo images[27,69]. Building on those existing methods, we developed a mathematical model that allows us to (1) determine the lengths of stereocilia tilted in any direction and (2) determine the lengths of stereocilia in which their stereocilia bases (insertion point into the cuticular plate) is obstructed by front row stereocilia. A detailed description can be found in the Supplementary Fig. 5.

**Statistics and reproducibility.** For statistical analysis, GraphPad Prism (La Jolla, CA), Excel, Matlab, and R were used. Two-way ANOVA was used to determine statistically significant differences in the ABR, DPOAE, and electrophysiology rise kinetic analyses. For two-tailed unpaired Student's $t$ test, $p$ values smaller than 0.05 were considered statistically significant, other values were considered not significant (n.s.). Asterisks in the figures indicate $p$ values (*$p < 0.05$, **$p < 0.01$, ***$p < 0.001$, and ****$p < 0.0001$). All error bars indicate standard deviation (SD) or standard error of the mean, as indicated in the figure legends.

Reproducibility of micrographs: Fig. 2a: IF images are representative of >20 experiments (independently performed in different animals). Figure 2b, c: IF images are representative of >5 experiments. Figure 3d: IF images are representative of >10 experiments. Figure 3f: fluorescence images are representative of three experiments. Figure 4e: IF image is representative of four experiments. Figure 5a: IF images are representative of four experiments. Figure 5b, d: IF images are representative of four experiments. Figure 8e: these SEM images are representative of five experiments. Supplementary Fig. S2: IF images are representative of four experiments.

*Electrophysiology*: All N's presented refer to the number of individual cells, with the number of animals used also indicated. Each current and displacement trace is the average of four presentations of the same stimulus intensity (technical replicates).

**Hearing tests in mice.** ABRs of WT and *Myo7a-ΔC* mice at P17, 4 week, 6 week, and 9 week were determined. Mice were anesthetized with a single intraperitoneal injection of 100 mg/kg ketamine hydrochloride (Fort Dodge Animal Health) and 10 mg/kg xylazine hydrochloride (Lloyd Laboratories). ABR and DPOAE were performed in a sound-attenuating cubicle (Med-Associates, product number: ENV-022MD-WF), and mice were kept on a Deltaphase isothermal heating pad (Braintree Scientific) to maintain body temperature. ABR recording equipment was purchased from Intelligent Hearing Systems (Miami, Fl). Recordings were captured by subdermal needle electrodes (FE-7; Grass Technologies). The noninverting electrode was placed at the vertex of the midline, the inverting electrode over the

mastoid of the right ear, and the ground electrode on the upper thigh. Stimulus tones (pure tones) were presented at a rate of 21.1/s through a high-frequency transducer (Intelligent Hearing Systems). Responses were filtered at 300–3000 Hz and threshold levels were determined from 1024 stimulus presentations at 8, 11.3, 16, 22.4, and 32 kHz. Stimulus intensity was decreased in 5–10 dB steps until a response waveform could no longer be identified. Stimulus intensity was then increased in 5 dB steps until a waveform could again be identified. If a waveform could not be identified at the maximum output of the transducer, a value of 5 dB was added to the maximum output as the threshold.

DPOAEs of same group of WT and *Myo7a-ΔC* mice were recorded. While under anesthesia for ABR testing, DPOAE were recorded using SmartOAE ver. 5.20 (Intelligent Hearing Systems). A range of pure tones from 8 to 32 kHz (16 sweeps) was used to obtain the DPOAE for right ear. DPOAE recordings were made for *f*2 frequencies from 8.8 to 35.3 kHz using paradigm set as follows: L1 = 65 dB, L2 = 55 dB SPL, and f2/f1 = 1.22.

**Whole-cell voltage-clamp electrophysiology recordings**. *Animals:* For electrophysiology and hair bundle motion recordings, C57BL/6J wild type and *Myo7a-ΔC* mutant mice of both sexes were used for experiments. All experiments were performed on mice aged postnatal day (P) 7–8. *Myo7a-ΔC* animals were bred in-house at 12/12 h light/dark cycle and had access to food and water ad libitum.

*Preparation and recordings*: Animals were euthanized by decapitation using methods approved by the University of Colorado IACUC. Organs of Corti were acutely dissected from P7–8 mice and placed in recording chambers as previously described[70]. Tissue was viewed using a 100× (1.0 NA, Olympus or 1.1 NA, Nikon) water-immersion objective with a Phantom Miro 320s or Veo 410L (Vision Research) camera on a Slicescope (Scientifica) or FN1 (Nikon) illuminated with a TLED+ 525 nm LED (Sutter Instruments). Tissue was dissected and perfused with extracellular solution containing (in mM): 140 NaCl, 2 KCl, 2 CaCl$_2$, 2 MgCl$_2$, 10 HEPES, 2 creatine monohydrate, 2 Na-pyruvate, 2 ascorbic acid, 6 dextrose, pH = 7.4, 300–310 mOsm. In addition, an apical perfusion, with pipettes tip sizes of 150–300 µm, provided local perfusion to the hair bundles.

*Electrophysiological recordings:* Whole-cell patch-clamp was achieved on IHCs from middle to apical cochlear turns or basal turn OHCs using an Axon 200B or Multiclamp 700B amplifier (Molecular Devices) with thick-walled borosilicate patch pipettes (2–6 MΩ) filled with an intracellular solution containing (in mM): 92 CsCl, 3.5 MgCl$_2$, 5 ATP, 5 creatine phosphate, 10 HEPES, 10 cesium BAPTA, 3 ascorbic acid, pH = 7.2, 280–290 mOsm. Experiments were performed at 18–22 °C. Whole-cell currents were filtered at 10 kHz and sampled at 100 kHz using a USB-6356 or USB-6366 (National Instruments) controlled by jClamp (SciSoft Company). All experiments used −80 mV holding potential, uncorrected for liquid junction potential. Only cells with initially <200 A of leak current were kept for data analysis.

*Hair bundle stimulation and motion recording*: Hair bundles are stimulated with a custom 3D printed fluid jet driven by a piezoelectric disc bender (27 mm 4.6 kHz; Murata Electronics 7BB-27-4L0). Thin-wall borosilicate pipettes were pulled to tip diameters of 5–20 µm, filled with extracellular solution, and mounted in the fluid-jet stimulator. The piezo disc bender was driven by waveforms generated using jClamp and the signals were filtered with an eight-pole Bessel filter at 1 kHz (L8L 90PF, Frequency Devices Inc.) or 2 kHz (3382, Krohn-Hite Corporation) and variably attenuated (PA5, Tucker Davis) before being sent to a high voltage/high current amplifier (Crawford Amplifier) to drive the piezoelectric disc. During stimulations, videos were taken of the hair bundle motion with high-speed imaging at 10,000 frames per second using the Phantom Miro 320s or Veo 410L. Videos were saved for each stimulation and analyzed offline.

*Analysis:* All data were analyzed offline using jClamp (SciSoft), Matlab (MathWorks), Excel (Microsoft), and Prism 7 (GraphPad). Figures were generated using Matlab, Prism 7, and Adobe Illustrator.

*Hair bundle motion analysis*: Custom Matlab (Mathworks) scripts were used for analysis of the hair bundle motion. Video frames were imported into Matlab and the position of the hair bundle was extracted using a Gaussian fit to a band-pass filtered hair bundle image[71] for a given vertical row of pixels in the image to yield subpixel resolution. For 50 ms fluid-jet stimulus steps, the motion of the hair bundle was fit with a double exponential, with the onset of the fit occurring after the force of the fluid-jet stimulus plateaus

$$y = y_0 + A_1 e^{-(x-x_0)/\tau_1} + A_2 e^{-(x-x_0)/\tau_2}, \qquad (1)$$

where $\tau_1$ and $\tau_2$ are the decay constants and $A_1$ and $A_2$ are the respective amplitudes. For a majority of cells, the center of the hair bundle was used to analyze motion.

*Electrophysiological data analysis:* The resting open probability of MET channels (defined as $P_{open}$, or $P_o$) was determined from sine wave stimulation of the hair bundle and calculated using Eq. (2), where $I_{max}$ is the current elicited during maximum positive stimulation, $I_{leak}$ is the current remaining during maximum negative stimulation, and $I_{resting}$ is the resting mechanosensitive current (defined as the current in the absence of stimulation, subtracted by $I_{leak}$). We assumed we could observe a $P_o$ of 0% and 100% during the maximum negative and

positive stimulations, respectively,

$$P_o = \frac{I_{resting}}{I_{max} - I_{leak}}. \qquad (2)$$

Current–displacement (IX) plots used the displacement data from the high-speed imaging analysis. To generate the IX plot, displacement values were taken when the maximum current occurred for each 50 ms step in a 13-step family of stimulations. Normalized currents ($I/I_{max}$) were generated by subtracting $I_{leak}$ and normalizing to the maximum current recorded for the entire stimulus family, which typically occurred on the 12th or 13th stimulus. IX plots were fit with a double Boltzmann equation

$$y = \frac{I_{max}}{1 + e^{Z_2(x_0-x)}\left(1 + e^{Z_1(x_0-x)}\right)}, \qquad (3)$$

where $Z_1$ and $Z_2$ are the slope factors and $x_o$ is the set point.

*Analysis of MET current and hair bundle rise kinetics:* For analysis of the MET current and hair bundle displacement rise kinetics, the time for 10 and 90% of the maximum motion or current to rise for each 50 ms stimulus step was determined from the raw displacement data obtained from the high-speed imaging, or the raw MET current data. In the text and figures, $t_{10\%}$ and $t_{90\%}$ refer to the time (in milliseconds) for 10% and 90% of the motion or current to rise, respectively. The $t_{10-90\%}$ is defined as the time for between 10 and 90% of the motion or current to rise, and is calculated by subtracting $t_{10\%}$ from $t_{90\%}$.

**Reporting summary**. Further information on research design is available in the Nature Research Reporting Summary linked to this article.

## Data availability
All relevant data are available from the authors. The source data for all figures are included in the Source Data file.

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

## Acknowledgements

We would like to thank Ulrich Mueller and Dominic Cosgrove for providing the antibodies against harmonin and ADGRV1, respectively. We also would like to thank Drs Jong-Hoon Nam and Jeff Corwin for helpful discussions on the manuscript. This study was supported by NIH/NIDCD grants R01DC014254 and R56DC017724 (to J.-B.S.), R01DC016868 (to A.W.P.), and by the National Institute on Deafness and Other Communication Disorders Intramural Research Program Z01-DC000002 (to B.K.)

## Author contributions

S.L., A.M., J.K., E.L.W., T.-T.D., G.A.C., R.C., I.T.R., B.K., A.W.P., and J.-B.S. performed the experiments. S.L., A.M., J.K., G.A.C., B.K., A.W.P., and J.-B.S. analyzed the data. L.P. helped in developing the mathematical model to measure stereocilia length. W.X. helped to generate the mouse models. S.L., A.M., A.W.P., and J.-B.S. wrote the manuscript. A.W.P. and J.-B.S. supervised this project.

## Competing interests

The authors declare no competing interests.
