## [Peer Review File · Nature Communications]

Reviewers' comments:

Reviewer #1 (Remarks to the Author):

In this interesting study Li and colleagues report on a detailed study of myosin VIIa (myo7a) function in mechanotransduction in mouse hair cells. They find two isoforms in the mouse cochlea and analysed the role of the longer isoform (myo7a-l) in an isoform-specific knock-out mouse myo7a-delta-l. The authors argue based on analysis of expression and function that this disruption primarily affects the inner hair cells where the myo7a expression was markedly reduced. Different from previously described myo7a mouse mutants, hair bundle morphology was found to be better maintained in myo7a-delta-l mice. Using hair cell physiology a rather subtle but interesting but specific alteration of mechanotransduction: markedly reduced resting open probability, which seems consistent with the previously stated hypothesis that myo7a works in generating sufficient tip-link tension by moving the upper tip-link density towards the tip of the stereocilia. The systems physiology analysis involved recordings of auditory brainstem responses (ABRs) and of otoacoustic emissions and suggest normal function of outer hair cells and initially near normal auditory cochlear function followed by a progressive hearing impairment. The latter was attributed to degeneration of inner hair cell hair bundles as indicated by ultrastructural analysis of cochleae in aged mice.

Data is interesting, methods are overall sound (see some scepticism in the detailed comments), paper is written and figures prepared well.

There are a few issues that need to be addressed for this work to be published here. For example, ABRs need age-matched controls.

Interpretation of the phenotype is not trivial and the authors suggest that it is co-determined by partial compensation of myo7a-l loss by the myo7a-short isoform.

Intro: not sure about the sentence: "Importantly, this cell-type specific expression of distinct MYO7A isoforms, varying along the tonotopic gradient, suggests a potential mechanism by which hair cells tune tip-link tension and resting Po."

Generally, there is efforts to quantify the changes in myo7a expression and hair bundle morphology. However, statements like "WT-like bundle morphology at P5, as analyzed by both immunofluorescence and scanning electron microscopy" are supported by exemplary Phalloidin stains and SEM and could be further substantiated by quantification.

Figure 2, legend: "MYO7A immunoreactivity is significantly reduced in all utricle hair cells" in the absence of statistical evaluation the term "significantly" should not be used.

Moreover, it is essential that the authors work on closing the gap between the majority of morphological and cellular physiology data that was acquired in neonatal cochleae and other aspects of the study obtained from older mice. At the least the MS should communicate this and in particular when interpreting the data.

The ratiometric analysis of myo7/6 immunofluorescence is a strong point. Yet, the authors want to either present/quote evidence for the absence of myo6 tonotopic gradients or be more cautious in interpretation.

The myo7a-promoter driven actin-GFP mouse transgenesis experiment is also very nice (also to validate the notion of preferential expression myo7a-l in IHCs. From the exemplary whole-mounts in Fig 3d it seems as if the IHC expression follows a tonotopic gradient which is not obvious from the myo7a analysis in wt IHC (fig 3a/b). Is my impression from fig 3d supported by quantitative data the authors might have and how can it be reconciled with 3a/b?

Analysis of myo7a immunofluorescence at upper tip-link density and taper of stereocilia:

The demonstration of residual myo7a abundance at the upper tip-link density is important.

However, there are some aspects that the authors might want to consider

- i) the number of spots seems much lower for the myo7a-l mut and one wonders whether this is because there were fewer upper tip-link densities with detectable myo7a. If this was likely I would consider an analysis of the number of myo7a-deficient upper tip-link densities very important. Ideally this would involve co-staining with a upper tip-link density marker such as harmonin.
2. if the number is indeed lower in the mutant hair cells, it might be good to consider and discuss the possibility that the Myo7a immunofluorescence per spot in reality is still lower in the mutant, because analysis missed on weaker spots
3. how was the normalization performed, where was the background collected: please ideally indicate the background area on graph

MET:

Please provide a short statement of how open probability was estimated in main text or legend to fig 4.

"Using 0.1mM BAPTA intracellular concentration, which represents a more physiological calcium buffering capacity ref 36," this statement seem unreasonable and the authors might want to consult Pangrsic et al., PNAS 2015 for a functional estimation of inner hair cell Ca²⁺ buffering

The notion of a reduced resting open probability seems incompatible with not finding a significant shift in the current-displacement curve.

MET analysis by fluid jet stimulation reduces my enthusiasm for the paper as kinetics of the MET current activation cannot be resolved by the method. In my view, this is unfortunate and at the least the discussion should be toned down to reflect this weakness of the MS.

Fig. S1: "Arrowhead points to the onset of the exponential fit, which occurs after the force of the fluid jet stimulus has plateaued." I am confused by this attempt to analyse the mechanical properties of the hair bundle/stereocilia:

What is the evidence for the plateau of the force at this point? What is meant by onset of exponential: did the authors define the fitting range like this? How could the point of onset along y₀ be the same WT and Mut if then the relaxing components are larger in the Mut.

ABR: probably more meaningful to express the threshold increase rather than absolute thresholds (at least if the reference to wt is not provided), since the thresholds of wt mice usually get worse over age. It seems unreasonable to only provide the thresholds of 4-week-old wt.

The discussion of the new myo7a-l mutant in comparison to the conditional myo7a allele should be balanced better: you have worked in the presence of a myo7a background and this was likely also the case in the conditional experiments.

Reviewer #3 (Remarks to the Author):

This manuscript re-visits the role of unconventional myosin VIIa in setting up the resting tension

within mechano-electrical transduction (MET) machinery of the mammalian auditory hair cells. Despite some recently found controversies on the earlier reports, the role of myosin VIIa in MET tension has been established by several lines of evidence. These evidence include direct measurements of MET currents in the whole body or hair cell-specific myosin VIIa mutants (Kros et al., 2002; Corns et al., 2018) as well as the lack of FM1-43 or aminoglycoside influx through the MET channels in myosin VIIa-deficient hair cells (Gale et al., 2001; Richardson et al., 1997). However, an interesting feature of the current study is the finding that mammalian hair cells express at least two different isoforms of myosin VIIa. The authors generated a novel isoform-specific mouse model lacking only the longer isoform of myosin VIIa (Myo7a-DL). Characterization of this mouse model revealed unexpected differences in the expression of myosin VIIa isoforms in the inner (IHCs) and outer (OHCs) hair cells, which might explain functional differences in the MET apparatus of these cells. A weakness of the study is that the data do not provide evidence for direct involvement of myosin VIIa in MET tension, contrary to the major claim of the manuscript. Furthermore, essential controls are lacking and the study does not explore functional differences between IHCs and OHCs in Myo7a-DL mice, which would have greatly strengthened the paper.

Specific comments are as follows:

The hair bundle is a mechanical system where the tension of the MET apparatus at rest is counter balanced by the pivot stiffness of the bundle's structure. This is evidenced, for example, by the positive deflection of the bundle after disruption of the tip links (Assad et al., 1991). Therefore, the resting tension experienced by the MET channels could be changed equally by both: the decrease of tip link tension and/or decrease of the bundle's stiffness. The data (Fig. 4a-d) clearly show significantly larger decrease of Myo7a expression at the base of IHC stereocilia in Myo7a-DL mice as compared to the region of the upper tip link density (UTLD). Furthermore, the data also indicate the potential changes of the pivot stiffness of the IHC bundle (Supplemental Figure 1). Yet, the authors consider mechanical changes at the base of IHC stereocilia only hypothetically. Certainly, measurements of the bundle stiffness, as well as ultrastructural examination of the base of IHC bundle (including ankle-links) is required to validate the major conclusion of the manuscript.

Difference in the expression of Myo7a isoforms in IHCs and OHCs (Fig. 3b) is probably the most exciting finding of this study. Besides MET abnormalities in IHCs, it predicts perfectly normal transduction in basal OHCs and some potential deficiencies in apical OHCs. Yet, MET currents in the OHCs of Myo7a-DL mutants were not investigated at all.

Likewise, the reader has to believe that OHCs in Myo7a-DL mice have normal stereocilia bundle structure based only on two hand-picked SEM images (Fig. 6c). This is nowhere close to the extensive analysis of the IHCs in Fig. 6d-f. Again, the differences between OHCs and IHCs seem to be the major result and it needs to be quantified not only for mutant IHCs but also for mutant OHCs, since OHC phenotype may be very subtle.

Figure 4: Would it be possible to provide TEM image of UTLD in Myo7a-DL IHCs to confirm that myosin VIIa deficiency does not disrupt UTLD?

Figure 5f: Why would negative displacements of the bundle not close MET channels even in the wild type?

Figure 6b: Where is the noise floor for DPOAE measurements?

Author's response

We would like to thank the reviewers for the opportunity to address the issues raised in their comments. We have conducted a significant body of additional experiments to address most concerns, including but not limited to an extensive quantification of hair bundle morphology at different ages, immunostaining for components of the upper tip link density and the ankle links, and additional electrophysiology experiments in cochlear inner and outer hair cells. We are happy to present a much improved manuscript. In the revised manuscript, we highlighted significant additions and modifications in **blue font**. For better visibility, the numerous minor changes (typos, minor language and content edits etc) are not highlighted.

We changed the title of the manuscript from initially “Myosin VIIa generates tension on the hair cell mechanotransduction complex” to **Myosin-VIIa is expressed in multiple isoforms and essential for tensioning the hair cell mechanotransduction complex**. The new title better captures the two crucial findings of our study.

General comment about the naming of MYO7A isoforms in the manuscript: We had initially named the two MYO7A isoforms MYO7A-L (for long isoform) and MYO7A-S (for short isoform). We believe that MYO7A-C (for canonical MYO7A isoform) is a more appropriate name for MYO7A-L. Text and figures were updated accordingly with the new name.

Most significant data addition that was not prompted by reviewer's comments: A major conclusion of our study is that the canonical (long) MYO7A-C isoform is differentially expressed in inner and outer hair cells. Our initial manuscript however lacked direct evidence for this, instead inferred the expression pattern based on the isoform-specific deletion and *Myo7a* promoter analysis. Because this is a major pillar of our study, we decided to confirm this by generating a new mouse line, in which we knocked-in the HA tag specifically into the canonical (long) MYO7A isoform. The expression of the tagged MYO7A-C is in full agreement with the predicted expression pattern.

Please find below a point-by-point address of the critique points.

Reviewer #1:

In this interesting study Li and colleagues report on a detailed study of myosin VIIa (myo7a) function in mechanotransduction in mouse hair cells. They find two isoforms in the mouse cochlea and analysed the role of the longer isoform (myo7a-l) in an isoform-specific knock-out mouse myo7a-delta-l. The authors argue based on analysis of expression and function that this disruption primarily affects the inner hair cells where the myo7a expression was markedly reduced. Different from previously described myo7a mouse mutants, hair bundle morphology was found to be better maintained in myo7a-delta-l mice. Using hair cell physiology a rather subtle but interesting but specific alteration of mechanotransduction: markedly reduced resting open probability, which seems consistent with the previously stated hypothesis that myo7a works in generating sufficient tip-link tension by moving the upper tip-link density towards the tip of the

stereocilia. The systems physiology analysis involved recordings of auditory brainstem responses (ABRs) and of otoacoustic emissions and suggest normal function of outer hair cells and initially near normal auditory cochlear function followed by a progressive hearing impairment. The latter was attributed to degeneration of inner hair cell hair bundles as indicated by ultrastructural analysis of cochleae in aged mice.

Data is interesting, methods are overall sound (see some scepticism in the detailed comments), paper is written and figures prepared well.

There are a few issues that need to be addressed for this work to be published here. For example, ABRs need age-matched controls.

We now display the ABR data as relative differences to age-matched controls in figure 7

Interpretation of the phenotype is not trivial and the authors suggest that it is co-determined by partial compensation of myo7a-l loss by the myo7a-short isoform.

We discuss this now in more detail in the Discussion section:

“...In comparison, our Myo7a-ΔC mice enable experiments at any age and importantly, allow the investigation of the functional significance of MYO7A isoforms. The interpretation of the Myo7a-ΔC mouse phenotype needs to consider the fact that this mouse line is not a mere MYO7A hypomorph, but null for a specific isoform. It is therefore possible that some aspects of the described phenotype are caused by a combination of a depletion of overall MYO7A levels and the loss of isoform-specific functions. How exactly the two MYO7A isoforms, by virtue of their expression levels and/or intrinsic properties, contribute to hair cell physiology remains to be studied in the future....”

Intro: not sure about the sentence: “Importantly, this cell-type specific expression of distinct MYO7A isoforms, varying along the tonotopic gradient, suggests a potential mechanism by which hair cells tune tip-link tension and resting Po.”

We acknowledge that this sentence is too speculative and misplaced in the introduction. We moved this topic to the Discussion.

Generally, there is efforts to quantify the changes in myo7a expression and hair bundle morphology. However, statements like “WT-like bundle morphology at P5, as analyzed by both immunofluorescence and scanning electron microscopy” are supported by exemplary Phalloidin stains and SEM and could be further substantiated by quantification.

Yes, we agree. The revised manuscript now contains an extensive quantification of stereocilia lengths at P7 (age of patch clamp experiments) and mature ages (age of hearing tests), in both inner and outer hair cells. This can be found in the results section and figures 2 and 7.

Figure 2, legend: “MYO7A immunoreactivity is significantly reduced in all utricle hair cells” in the absence of statistical evaluation the term “significantly” should not be used.

The quantification is now shown and the difference is indeed statistically significant. This is now stated in the results section and figure 2 legend.

Moreover, it is essential that the authors work on closing the gap between the majority of morphological and cellular physiology data that was acquired in neonatal cochleae and other aspects of the study obtained from older mice. At the least the MS should communicate this and in particular when interpreting the data.

Yes, we agreed with the concern. We believe that the extensive quantification of hair bundle morphology at P7 (age when cellular patch clamp physiology was performed) and mature ages (age of functional hearing tests) partly closes this gap, by providing an age-matched correlation between morphology and MET function (at P7), and morphology and ABR thresholds (in older mice). Our data is inconclusive in whether in older mutant mice, a direct causal connection can be made between MET deficit, bundle morphology deficits and hearing loss, and we thank the reviewer for pointing this out. We have now included a short paragraph to address this issue in the Discussion:

“Myo7a-ΔC mice develop a progressive form of hearing loss. Based on the mild MET current phenotype in Myo7a-ΔC IHCs, it is not surprising that hearing thresholds are not severely affected in wean age animals. With increasing age, MYO7A hypofunction might lead to significant MET dysfunction and/or broad defects in cellular homeostasis, both of which might affect hair bundle morphology and hearing. One must be cautious however to interpret the adult hearing loss phenotype solely from the angle of hair cell MET. MYO7A is widely localized in the hair cell, suggesting involvement in a range of yet unknown cellular processes. It is therefore possible that MYO7A hypofunction causes cellular dysfunction that are MET-independent, and that those deficits contribute to the progressive hearing loss.”

The ratiometric analysis of myo7/6 immunofluorescence is a strong point. Yet, the authors want to either present/quote evidence for the absence of myo6 tonotopic gradients or be more cautious in interpretation.

Yes, this is a valid point. We have now quantified MYO6 expression in cochlear hair cells in supplementary figure 1. Not only is MYO6 expression invariant along the tonotopic gradient, but there is also no significant change of MYO6 expression levels in the *Myo7a-ΔC* mice compared to WT.

The myo7a-promoter driven actin-GFP mouse transgenesis experiment is also very nice (also to validate the notion of preferential expression myo7a-I in IHCs. From the exemplary whole-mounts in Fig 3d it seems as if the IHC expression follows a tonotopic gradient which is not obvious from the myo7a analysis in wt IHC (fig 3a/b). Is my impression from fig 3d supported by quantitative data the authors might have and how can it be reconciled with 3a/b?

Yes, we agree that the *Myo7a::Actin-GFP* expression as shown in the initial submission appeared to show such a gradient. Displaying both IHCs and OHCs with vastly different Actin-GFP signals and stereocilia lengths was challenging, causing saturation in IHCs and barely visible signals in OHCs. In the present version, the cuticular plate regions of each panel exhibit a similar level of Actin-GFP signal, providing a reference point for a better comparison of the Actin-GFP expression levels at the different tonotopic positions.

Analysis of myo7a immunofluorescence at upper tip-link density and taper of stereocilia: The demonstration of residual myo7a abundance at the upper tip-link density is important.

However, there are some aspects that the authors might want to consider i) the number of spots seems much lower for the myo7a-l mut and one wonders whether this is because there were fewer upper tip-link densities with detectable myo7a. If this was likely I would consider an analysis of the number of myo7a-deficient upper tip-link densities very important. Ideally this would involve co-staining with a upper tip-link density marker such as harmonin.

If the number is indeed lower in the mutant hair cells, it might be good to consider and discuss the possibility that the Myo7a immunofluorescence per spot in reality is still lower in the mutant, because analysis missed on weaker spots

Yes, the reviewer points out an important question: does the reduction of MYO7A possibly affect the formation or abundance of the upper tip-link densities? As suggested, co-staining for MYO7A and harmonin would be the ideal way to investigate this, but because the antibodies for MYO7A and harmonin suitable for upper tip-link density detection are both rabbit antibodies, we could not perform such co-staining. We therefore performed separate immunostainings for harmonin and showed that harmonin intensities at the presumed upper tip-link densities is not reduced (even slightly increased) in *Myo7a-ΔC* IHCs (new Figure 5). Furthermore, the average number of harmonin puncta per cell was not significantly different between *Myo7a-ΔC* IHCs and WT counterparts. We interpret this result to mean that the residual amount of MYO7A (~15% of WT levels) is sufficient to mediate the normal formation of the upper tip-link densities.

3. how was the normalization performed, where was the background collected: please ideally indicate the background area on graph

The fluorescence intensity of MYO7A puncta was normalized against a dark background surrounding the UTLD staining in the same image and z-section. This is now noted in the figure legend.

MET:

Please provide a short statement of how open probability was estimated in main text or legend to fig 4.

Resting MET current was estimated using the difference in the resting current before stimulation minus the current with a maximal negative stimulation. The maximum MET current was estimated by the peak positive MET current minus the peak negative MET current. The resting open probability was the resting MET current divided by the maximum MET current. This is described in the Methods section and now also in the figure legend.

“Using 0.1mM BAPTA intracellular concentration, which represents a more physiological calcium buffering capacity ref 36,” this statement seem unreasonable and the authors

might want to consult Pangrsic et al., PNAS 2015 for a functional estimation of inner hair cell Ca²⁺ buffering.

The reviewer brings up a good point that the physiological buffer estimations can differ depending on how it is estimated. We have removed this statement and present only data with 10 mM BAPTA to be consistent with the sine wave stimulations.

The notion of a reduced resting open probability seems incompatible with not finding a significant shift in the current-displacement curve.

Yes, we agree with the reviewer on this. With 0.1mM BAPTA the trend was there but did not reach significance. We believe this is due to the lower resting tension (lower resting P_o), which makes the difference between KO and WT more difficult to resolve. For greater resolution and separation, we now use data for IX curves obtained with 10 mM BAPTA (higher resting P_o) and confirm that there is a significant shift in the IX curve as expected.

MET analysis by fluid jet stimulation reduces my enthusiasm for the paper as kinetics of the MET current activation cannot be resolved by the method. In my view, this is unfortunate and at the least the discussion should be toned down to reflect this weakness of the MS.

We understand the reviewers concern with this, however, even with stiff probe stimulation, the true activation kinetics of WT cochlear hair cells cannot be resolved, since the current activation is as fast as cell can be stimulated (see Grillet et al, 2009) . Our use of a slower stimulus does not diminish the fact that the KO is delayed. If we had a faster stimulus, this difference could be even greater. In addition, stiff probe stimulation has significant drawbacks in this type of experiment for IHCs. IHCs possess a near linear row of long stereocilia, are ~8 μm wide, which requires that the probe be 8 μm wide to engage all the stereocilia. However, due to the fabrication of the glass probes, which requires that the probe ends be round, an 8 μm wide probe has to be 8 μm tall as well. This geometry of the probe can unknowingly engage the shorter rows of stereocilia or the epithelial surface when stimulating the hair bundle. Additionally, when the probe is placed on the hair bundle, any bias the probe puts on the hair bundle can change the apparent resting P_o. This pre-tensing of the tip-links can affect the activation kinetics. When taken together with the larger geometry of the probe, even when the probe may not be touching the tallest row visually, it can be pushing on shorter rows of stereocilia and affecting resting P_o and activation kinetics. These concerns are much more relevant with IHCs than OHCs, which were used previously to investigate activation kinetics in the harmonin mutants (Grillet, et al, 2009). On the other hand, the fluid jet does engage all stereocilia (although we do acknowledge potential complex fluid dynamics issues) and is not biased by probe positioning since it does not require touching the hair bundle. And while the fluid jet is slower, it was in this case fast enough to dissect differences in current onset between the WT and *Myo7a-ΔC* IHCs. We have added some of these justifications into the results section.

Fig. S1: “Arrowhead points to the onset of the exponential fit, which occurs after the force of the fluid jet stimulus has plateaued.” I am confused by this attempt to analyse the mechanical properties of the hair bundle/stereocilia: What is the evidence for the plateau of the force at this point? What is meant by onset of exponential: did the authors define the fitting range like this? How could the point of onset along y0 be the same WT and Mut if then the relaxing components are larger in the Mut.

We apologize for the oversight in not providing this information. The evidence that the force plateaus 0.5ms after the onset of the stimulus is from data that is now published (Caprara et al, 2019, J Neurosci). The onset of the exponential is where we observe the hair bundle creep (see Caprara et al., 2019). After 0.5ms we fit the hair bundle creep with a double exponential to describe the kinetics. The y0 point is where the exponential plateaus, not the onset of where the exponential fit starts. So, the same y0 suggests that the overall displacement size of the WT and mutant are the same, but the relative proportion between the creep magnitude to the total displacement (quantified as $(A1 + A2)/y0$) is different. We are trying to be transparent in the findings that we have and providing this data since there are some differences in the mechanics of the hair bundle. However, what these differences mean at this point is difficult to interpret, since we do not know the origins of the hair bundle creep. We have clarified the description of the now figure S2 and put in the required reference.

ABR: probably more meaningful to express the threshold increase rather than absolute thresholds (at least if the reference to wt is not provided), since the thresholds of wt mice usually get worse over age. It seems unreasonable to only provide the thresholds of

4-week-old

wt.

To reduce the complexity and improve data visibility of the ABR threshold plots, we had placed the WT thresholds for each age in a Supplementary figure, but we agree with the reviewer that displaying the ABR data as relative difference to age-matched controls is the better option. The figure was revised accordingly.

The discussion of the new myo7a-l mutant in comparison to the conditional myo7a allele should be balanced better: you have worked in the presence of a myo7a background and this was likely also the case in the conditional experiments.

Yes, we agree with the reviewer in this. This is also related to the reviewer's previous comment about the non-trivial interpretation of the isoform-specific KO. The new paragraph reads as follows:

“Our present study is consistent with a more recent study that used a conditional knockout approach to study MYO7A's role in MET in intact hair cells (Corns et al., 2018). In $Myo7a^{fl/fl}Myo15-Cre^{+/-}$ mice, the Myo15 promoter drives Cre-mediated Myo7a gene deletion in hair cells starting around P4, allowing hair cells to develop normally. MYO7A levels presumably reached critically low levels after P14 in IHCs, at which point electrophysiological recordings revealed a reduction in resting P_o , consistent with our results. Due to slow protein turnover after genetic deletion, functional experiments on MYO7A-depleted hair cells were limited to mice aged P14 and older. In comparison, our $Myo7a-\Delta C$ mice enable experiments at any age and importantly, allow the investigation

of the functional significance of MYO7A isoforms. The interpretation of the *Myo7a-ΔC* mouse phenotype needs to consider the fact that this mouse line is not a mere MYO7A hypomorph, but null for a specific isoform. It is therefore possible that some aspects of the described phenotype are caused by a combination of a depletion of overall MYO7A levels and the loss of isoform-specific functions. How exactly the two MYO7A isoforms, by virtue of their expression levels and/or intrinsic properties, contribute to hair cell physiology remains to be studied in the future. In summary, the conditional and isoform-specific KO mouse models are complementary tools in studying MYO7A's role in hair cell function.”

Reviewer #3 (Remarks to the Author):

This manuscript re-visits the role of unconventional myosin VIIa in setting up the resting tension within mechano-electrical transduction (MET) machinery of the mammalian auditory hair cells. Despite some recently found controversies on the earlier reports, the role of myosin VIIa in MET tension has been established by several lines of evidence. These evidence include direct measurements of MET currents in the whole body or hair cell-specific myosin VIIa mutants (Kros et al., 2002; Corns et al., 2018) as well as the lack of FM1-43 or aminoglycoside influx through the MET channels in myosin VIIa-deficient hair cells (Gale et al., 2001; Richardson et al., 1997).

The reviewer references a list of previous studies to suggest that the role of Myo7a in MET tension is well established. We would like to make an argument that most of the referenced studies are not suited to draw any conclusion about Myo7a's role in tip-link tensioning, and that therefore the molecular identity of the tip link tensioner is still unknown. Previous work that supported the hypothesis that Myo7a generates tension in the tip-link (Kros et al., 2002; Gale et al., 2001; Richardson et al., 1997) requires re-interpretation because all of these studies used the Myo7a (6J/6J) mutant mouse line. Any results regarding MET resulting from these KO mice cannot be interpreted to be a result of a direct effect of Myo7a on MET. As the reviewer mentioned, there is controversy surrounding the results interpreted as effect on MET with this KO line. In Marcotti et al (2014, J Neurosci), the authors write in the discussion: “Our present findings that the anomalous MET currents in the OHCs of homozygous *Myo7a*^{6J} mutant mice are resistant to BAPTA treatment prompts a reinterpretation of the findings reported by Kros et al. (2002). We must now conclude that the MET currents in these mutants are not gated by means of tip links and are predominantly of opposite polarity relative to normal MET currents.” This indicates that the “MET current” recorded in these mice is not due to the MET channels that are measured in WT mice, but rather piezo2 channels (Wu et al, 2017, Nat Neurosci). Therefore, the machinery of the normal transduction is not functional in these disorganized hair bundles and the lowered resting open probability found in other studies referenced by the reviewer in the *Myo7a*^{6J} mice is not a measure or indication of effects on the normal mechanotransduction process. The argument that the *Myo7a*^{6J} mice do not have normal mechanotransduction, therefore Myo7a is involved in tensioning the MET complex is weak, since the hair bundles are highly disorganized and it is highly plausible that the MET machinery never

makes it to the required location. These developmental defects in *Myo7a*^{6J} mice are the crux of our argument that the *Myo7a-ΔC* mice, which do not have developmental defects, provide a unique opportunity to investigate the functional role of Myo7a in MET. The reviewer is correct that in the Corns et al. (2018) paper, the authors do find lowered resting P_o in a late onset conditional KO, which is consistent with our results, and we cite and discuss this work in the discussion. However, the authors did not make the point of how Myo7a may be involved in tensioning the MET apparatus, and we provide further data supporting this role. Thus, we believe our work here provides significant functional evidence of Myo7a mediated tensioning of the MET complex.

However, an interesting feature of the current study is the finding that mammalian hair cells express at least two different isoforms of myosin VIIa. The authors generated a novel isoform-specific mouse model lacking only the longer isoform of myosin VIIa (Myo7a-DL). Characterization of this mouse model revealed unexpected differences in the expression of myosin VIIa isoforms in the inner (IHCs) and outer (OHCs) hair cells, which might explain functional differences in the MET apparatus of these cells.

A weakness of the study is that the data do not provide evidence for direct involvement of myosin VIIa in MET tension, contrary to the major claim of the manuscript. Furthermore, essential controls are lacking and the study does not explore functional differences between IHCs and OHCs in Myo7a-DL mice, which would have greatly strengthened the paper.

Specific comments are as follows:

The hair bundle is a mechanical system where the tension of the MET apparatus at rest is counter balanced by the pivot stiffness of the bundle's structure. This is evidenced, for example, by the positive deflection of the bundle after disruption of the tip links (Assad et al., 1991). Therefore, the resting tension experienced by the MET channels could be changed equally by both: the decrease of tip link tension and/or decrease of the bundle's stiffness. The data (Fig. 4a-d) clearly show significantly larger decrease of Myo7a expression at the base of IHC stereocilia in Myo7a-DL mice as compared to the region of the upper tip link density (UTLD). Furthermore, the data also indicate the potential changes of the pivot stiffness of the IHC bundle (Supplemental Figure 1). Yet, the authors consider mechanical changes at the base of IHC stereocilia only hypothetically. Certainly, measurements of the bundle stiffness, as well as ultrastructural examination of the base of IHC bundle (including ankle-links) is required to validate the major conclusion of the manuscript.

The reviewer is correct that resting tension in the hair bundle has to work against the stiffness of the hair bundle's structure, such that for a given resting tension, a higher stiffness of the hair bundle would result in a smaller movement in resting position when tip links are broken (the motion observed in Assad et al, 1991 and more recently in Tobin et al, 2019). However, the hair bundle stiffness affects only the movement in resting position due to tip-link breakage, not the actual resting tension generated by the motors. The motors provide a force, which is translated to a motion of the hair bundle based on $F=kx$, where F is the force, k is the stiffness of the hair bundle without tip links (the stiffness that the resting tension needs to work against), and x is the motion that is

observed when tip links are broken. So, the force is independent of the stiffness of the hair bundle. This is exemplified experimentally in the *Triobp* KO mice (Kitajiri et al., 2010). In these mice, the overall hair bundle stiffness is reduced by about 50%, but there is no change in the resting open probability of the MET channel, suggesting that resting tension in the tip link is unaltered. We would predict that in the *Triobp* KO mice that the movement associated with tip-link breakage would be larger than in WT, since $x = F/k$, where k is smaller in the *Triobp* KO hair bundles.

We believe this is an important discussion, and in the manuscript, we now devote a whole paragraph to this discussion:

“In our study, reduced resting P_o serves to indicate a less active tip-link motor. It is important to discuss the validity and robustness of this connection. In the *Myo7a-ΔC* IHCs, MYO7A is not only reduced at the UTLD, but also at the stereocilia base. Although we did not detect significant changes in the level or localization of the ankle link component ADGRV1³¹, it is possible that the stiffness of the stereocilia base, thus pivot stiffness, is affected in the *Myo7a-ΔC* IHCs. In fact, analysis of stereocilia motion in response to fluid jet stimuli suggest that the mechanical properties, and potentially stiffness of the hair bundle, are altered in *Myo7a-ΔC* mice. The crucial question is whether changes in such additional “springs” in the system might influence tip-link tension and resting P_o indirectly. A judgment on this depends on the interpretation of the existing hair cell MET model. According to a generally accepted assumption⁴⁷, the tip-link is tensed by a motor protein that pulls with a certain force (e.g., the term F_m in^{48–51}). This force is maintained (force-clamped) regardless of the stiffness of other springs in the system. Mechanistically, this might be achieved by the motor climbing up the actin ladder until its stall force is reached. Experimentally, this interpretation is supported by the phenotype of the *Triobp* KO mice, in which, even though the pivot stiffness is severely reduced, hair cell MET currents show no signs of reduced resting P_o (e.g. changes in the activation curve)⁵². Based on this reasoning, we suggest that resting P_o , regardless of changes in overall hair bundle stiffness, serves as a reasonably robust proxy for tip-link motor activity. The hair bundle however is a complex mechanical system, and present assumptions about the tip link motor might be simplistic. Direct mechanical measurements of tip-link tension, using previously described methods^{53,54}, will provide more conclusive data on whether the reduction in resting P_o in *Myo7a-ΔC* IHCs is indeed accompanied by reduced tip-link tension.”

In light of this discussion, we felt that hair bundle stiffness measurements were beyond the immediate scope of this study. As in the initial manuscript, we do describe the changes in hair bundle mechanics by analyzing the bundle relaxation (“creep”) in WT and the *Myo7a-ΔC* IHCs stimulated with a fluid jet, a method we used in a recent paper (Caprara et al., 2019). The “creep” was found to be greater in the *Myo7a-ΔC* IHCs. In the manuscript, we merely conclude that “The differences in motion suggest that the mechanical properties of the hair bundle are altered in *Myo7a-ΔC* mice.” We felt that the present data did not allow us to make more definitive mechanistic conclusions for example with regards to changes in stiffness.

TEM studies of the ankle link region were attempted, but the results were not conclusive in establishing whether the stereocilia base and roots are altered in the *Myo7a-ΔC* IHCs. We were able to visualize rootlets and stereocilia bases in both genotypes, but the frequency of visualizing these structures with TEM was low, therefore difficult to make quantitative comparisons. In lieu of a TEM study, we therefore examined the expression and localization of ADGRV1 at the ankle link region, which is known to require MYO7A for its formation. We did not detect any differences in the level or localization pattern of ADGRV1, suggesting that the ankle link is formed normally in *Myo7a-ΔC* IHCs. And again, even if the stereocilia pivot stiffness was reduced in *Myo7a-ΔC* mice, we predict that tip-link tension and resting open probability of the MET channel will not be affected, as evident in the *Triobp KO* mice.

Difference in the expression of Myo7a isoforms in IHCs and OHCs (Fig. 3b) is probably the most exciting finding of this study. Besides MET abnormalities in IHCs, it predicts perfectly normal transduction in basal OHCs and some potential deficiencies in apical OHCs. Yet, MET currents in the OHCs of Myo7a-DL mutants were not investigated at all.

We agree with the reviewer and have now performed experiments on OHCs in the base of the cochlea, which serves as an internal control to our data, since there are no significant differences in MYO7A levels found in basal OHCs. We found that basal OHCs did not differ in WT and KO animals for both sine wave and step responses. This data is now presented in Figure 6 and Supplementary figure 3.

Likewise, the reader has to believe that OHCs in Myo7a-DL mice have normal stereocilia bundle structure based only on two hand-picked SEM images (Fig. 6c). This is nowhere close to the extensive analysis of the IHCs in Fig. 6d-f. Again, the differences between OHCs and IHCs seem to be the major result and it needs to be quantified not only for mutant IHCs but also for mutant OHCs, since OHC phenotype may be very subtle.

We agree with the reviewer's comment. The revised manuscript now contains an extensive quantification of stereocilia lengths at P7 (age of patch clamp experiments) and mature ages (age of hearing tests) in figures 2 and 7.

Figure 4: Would it be possible to provide TEM image of UTLD in Myo7a-DL IHCs to confirm that myosin VIIa deficiency does not disrupt UTLD?

We have tried very hard, but in the end, we were not able to obtain TEM images of inner hair cell UTLDs, neither in WT nor in *Myo7a-ΔC*. It should be noted that most TEM images of UTLD in the literature are from OHCs, and that visualizing UTLD in IHCs is considered exceedingly difficult. We do agree with the reviewer however, that it is important to investigate whether the UTLD or the ankle link region are disrupted in the *Myo7a-ΔC* IHCs. This question was also raised by Reviewer #1. To address this, we performed immunostaining for harmonin (proxy for UTLD) and ADGRV1 (proxy for ankle link). Harmonin localization and fluorescence intensity was not reduced (even slightly increased), neither did we detect any changes in ADGRV1 in *Myo7a-ΔC* IHCs. This suggests that despite the reduced levels of MYO7A (and complete lack of

MYO7A-C), the formation of the UTLD and the ankle link complex is likely not affected in the *Myo7a-ΔC* IHCs. This is now shown in the new Figure 5.

Figure 5f: Why would negative displacements of the bundle not close MET channels even in the wild type?

There is a small current that is closed with negative displacements, however this small change is obscured by the variable baseline of the other MET traces (see below). We also now shade the most negative traces in the data with 10mM BAPTA presented in the paper for clarity.

Figure 6b: Where is the noise floor for DPOAE measurements?

We have added this to the DPOAE plot.

REVIEWERS' COMMENTS:

Reviewer #1 (Remarks to the Author):

The authors have performed a substantial revision, I endorse publication and wish to congratulate them on a very nice study.

There is one statement that they might to think about:

"In comparison, our Myo7a-ΔC mice enable experiments at any age and importantly, allow the investigation of the functional significance of MYO7A isoforms."

Since the patch-clamp was only done early postnatally in this study the statement kind of promises a bit more than they deliver.

Best
Tobias Moser

Reviewer #3 (Remarks to the Author):

In this revised version, the authors have added substantial amount of new data that significantly improved the manuscript. Particularly welcome are the new data on HA-tag expression in the new knock-in mice with tagged HA-Myo7a-C isoform, mechanotransduction recordings in basal OHCs, new data on harmonin and ADGRV1 localization, and quantification of both IHC and OHC bundles in Myo7a-C-deficient mice. Thus, the manuscript provides an impressive amount of data that will be valuable to the field.

However, with all respect, I cannot agree with the author's response to my major point that the manuscript data do not yet provide evidence for the DIRECT effect of MYO7A on the tensioning of the hair cell mechanotransduction (MET) complex. The author's argument that the resting tip link tension should not depend on the hair bundle stiffness simply contradicts the laws of physics. If you are sitting in the boat and pull the rope attached to the pier, the boat will start moving to the pier and the steady-state tension on the rope will be close to zero. Likewise, if myosin motors would pull up the upper ends of the tip links in a bundle with an indefinitely low stiffness, the taller stereocilia would start falling in negative direction without any significant tensioning of the tip links. The tip link tension would develop only after negative stereocilia deflection would meet some structural constrains (e.g. from ankle or other side links). By that time, the extent of the upper tip link density (UTLD) upward movement (which is extremely limited, see Grati and Kachar, PNAS, 2011) may be already exhausted. Therefore, the resting force in the MET complex would not depend on myosin's force and could be very low.

Similarly, the provided reference (Kitajiri et al., 2010) does not have any experimental data supporting the author's claim. That study used stiff probe stimulation for MET recordings, which does not reliably deflect the hair bundle in negative direction. Thus, it is not surprising that Kitajiri et al. never report any values of resting P_o in their paper. I am not sure how the authors concluded that "experimentally, this interpretation is supported by the phenotype of the Triobp KO mice, in which, even though the pivot stiffness is severely reduced, hair cell MET currents show no signs of reduced resting P_o (e.g. changes in the activation curve)". Perhaps, the authors were misled by overlapping $I(X)$ curves in panel C of figure 7 in Kitajiri et al., 2010. However, these curves don't show saturation of MET current on the negative bundle displacements and, therefore, cannot be used to determine P_o .

Having in mind the above considerations, I do like the latest change in the title that downplays MET tensioning and I would suggest the following edits in lines 372-380, "... This force is maintained (force-

clamped) regardless of the stiffness of other springs in the system. Mechanistically, this might be achieved by the motor climbing up the actin ladder until its stall force is reached. Based on this reasoning, we suggest that resting P_o MAY serve as a reasonably robust proxy for tip-link motor activity. The hair bundle however is a complex mechanical system, and present assumptions about the tip-link motor might be simplistic. ...”

Minor points remaining:

Figure 7A: It is very unusual to see the difference in ABR thresholds and not their absolute values. I know that this figure was modified in response to the comment of Reviewer 1, but the modification doesn't answer the comment directly. The only thing that was needed is to show how the thresholds are changing in age-matched WT controls. I would suggest two panels with absolute ABR thresholds (in dB SPL) side by side for WT and mutants. In contrast, the current version of Fig. 7A makes things more complicated. For example, since both WT and mutant measurements have errors, how the resulting error bars in Fig. 7A were determined?

Lines 397-399: I would be cautious in accepting a recent re-interpretation of the previous recordings of MET currents in *Myo7a*-deficient mice as an “abnormal” transduction driven by PIEZO2 channels (Marcotti et al., 2014). Unless been completely retracted, Fig.4c in the original paper of Kros et al. (2002) demonstrates strong MET responses only to positive deflections of the hair bundles at P6, when “abnormal” MET current should disappear (Beurg et al., PNAS, 2016). I would rather conclude that the previous studies were controversial but the new MET recordings in *Myo7a*-C-deficient mice provide a better insight into the role of MYO7A-C in transduction.

Lines 439-440: Please, provide reference for the following statement, “Second, a recent study in rats reported that tension in individual tip links is generally lower in IHCs compared to OHCs.” I guess, it is the ref. 55?

Author's response

We would like to thank you for the opportunity to respond to the final comments by the reviewers. In this documents, the text additions in response to the comments are highlighted in red and deleted text as ~~striketrough~~.

Responses to reviewer's comments:

Comment by Reviewer #1:

The authors have performed a substantial revision, I endorse publication and wish to congratulate them on a very nice study.

There is one statement that they might think about:

"In comparison, our *Myo7a-ΔC* mice enable experiments at any age and importantly, allow the investigation of the functional significance of MYO7A isoforms."

Since the patch-clamp was only done early postnatally in this study the statement kind of promises a bit more than they deliver.

Response:

We agree. We changed the sentence to: "In comparison, our *Myo7a-ΔC* mice enable experiments at ~~any age~~ **early postnatal ages** and importantly, allow the investigation of the functional significance of MYO7A isoforms."

Comments by Reviewer #3:

In this revised version, the authors have added substantial amount of new data that significantly improved the manuscript. Particularly welcome are the new data on HA-tag expression in the new knock-in mice with tagged HA-Myo7a-C isoform, mechanotransduction recordings in basal OHCs, new data on harmonin and ADGRV1 localization, and quantification of both IHC and OHC bundles in *Myo7a-C*-deficient mice. Thus, the manuscript provides an impressive amount of data that will be valuable to the field.

However, with all respect, I cannot agree with the author's response to my major point that the manuscript data do not yet provide evidence for the DIRECT effect of MYO7A on the tensioning of the hair cell mechanotransduction (MET) complex. The author's argument that the resting tip link tension should not depend on the hair bundle stiffness simply contradicts the laws of physics. If you are sitting in the boat and pull the rope attached to the pier, the boat will start moving to the pier and the steady-state tension on the rope will be close to zero. Likewise, if myosin motors would pull up the upper ends of the tip links in a bundle with an indefinitely low stiffness, the taller stereocilia would start falling in negative direction without any significant tensioning of the tip links. The tip link tension would develop only after negative stereocilia deflection would meet some structural constrains (e.g. from ankle or other side links). By that time, the extent of the upper tip link density (UTLD) upward movement (which is extremely limited, see Grati and Kachar, PNAS, 2011) may be already exhausted. Therefore, the resting force in the MET complex would not depend on myosin's force and could be very low.

Similarly, the provided reference (Kitajiri et al., 2010) does not have any experimental data supporting the author's claim. That study used stiff probe stimulation for MET recordings, which does not reliably deflect the hair bundle in negative direction. Thus, it is not surprising that Kitajiri et al. never report any values of resting P_o in their paper. I am not sure how the authors concluded that "experimentally, this interpretation is supported by the phenotype of the *Triobp*

KO mice, in which, even though the pivot stiffness is severely reduced, hair cell MET currents show no signs of reduced resting P_o (e.g. changes in the activation curve)". Perhaps, the authors were misled by overlapping $I(X)$ curves in panel C of figure 7 in Kitajiri et al., 2010. However, these curves don't show saturation of MET current on the negative bundle displacements and, therefore, cannot be used to determine P_o .

Having in mind the above considerations, I do like the latest change in the title that downplays MET tensioning and I would suggest the following edits in lines 372-380, "... This force is maintained (force-clamped) regardless of the stiffness of other springs in the system. Mechanistically, this might be achieved by the motor climbing up the actin ladder until its stall force is reached. Based on this reasoning, we suggest that resting P_o MAY serve as a reasonably robust proxy for tip-link motor activity. The hair bundle however is a complex mechanical system, and present assumptions about the tip-link motor might be simplistic. ..."

Response:

Although we generally stand by our conclusions, we agree that we could be a bit more cautious with our statements. We also would like to thank the reviewer for pointing out the possibility that the upwards mobility of the tip link motor complex might be a limiting factor for our "force-clamping" assumption. We therefore made the following changes:

"Experimentally, this interpretation **is appears to be indirectly** supported by the phenotype of the *Triobp* KO mice, in which, even though the pivot stiffness is severely reduced, hair cell MET currents showed no ~~signs of reduced resting P_o~~ **changes in the activation curve, suggestive of unchanged resting P_o** ⁵³. Based on this reasoning, we suggest that resting P_o , regardless of changes in overall hair bundle stiffness, **may** serve as a reasonably robust proxy for tip-link motor activity. The hair bundle however is a complex mechanical system, and present assumptions about the tip-link motor might be simplistic. **For example, the tip-link motor might not be able to adjust to severe reduction of the bundle stiffness if its upward mobility is limited, as was previously suggested**²¹. Direct mechanical measurements of tip-link tension, using previously described methods^{54,55}, will provide more conclusive data on whether the reduction in resting P_o in *Myo7a-ΔC* IHCs is indeed accompanied by reduced tip-link tension."

Minor points remaining:

Figure 7A: It is very unusual to see the difference in ABR thresholds and not their absolute values. I know that this figure was modified in response to the comment of Reviewer 1, but the modification doesn't answer the comment directly. The only thing that was needed is to show how the thresholds are changing in age-matched WT controls. I would suggest two panels with absolute ABR thresholds (in dB SPL) side by side for WT and mutants. In contrast, the current version of Fig. 7A makes things more complicated. For example, since both WT and mutant measurements have errors, how the resulting error bars in Fig. 7A were determined?

Response:

Yes, we agree with the reviewer on this and have modified the figure accordingly. In Fig. 7a, we now plot WT and mutant traces for each age.

Lines 397-399: I would be cautious in accepting a recent re-interpretation of the previous recordings of MET currents in *Myo7a*-deficient mice as an "abnormal" transduction driven by PIEZO2 channels (Marcotti et al., 2014). Unless been completely retracted, Fig.4c in the original paper of Kros et al. (2002) demonstrates strong MET responses only to positive deflections of

the hair bundles at P6, when “abnormal” MET current should disappear (Beurg et al., PNAS, 2016). I would rather conclude that the previous studies were controversial but the new MET recordings in Myo7a-C-deficient mice provide a better insight into the role of MYO7A-C in transduction.

Response:

In the manuscript Marcotti et al., 2014, the authors, including the author of the original Kros et al., 2002 manuscript, state:

“Our present finding that the anomalous MET currents in the OHCs of homozygous Myo7a6J mutant mice are resistant to BAPTA treatment (Fig. 4C) prompts a reinterpretation of the findings reported by Kros et al. (2002). We must now conclude that the MET currents in these mutants are not gated by means of tip links and are predominantly of opposite polarity relative to normal MET currents.”

The reviewer also noted that it was positive deflections that elicited the current in the Myo7a mutant in the Kros et al., 2002. However, in the Kros et al, 2002 manuscript they defined positive deflections as anything that elicited an inward current, and this is clarified in the Marcotti et al., 2014 Methods section:

“The fluid jet was positioned at the modiolar side of the hair bundles and positive driver voltage (fluid flowing out of the jet) corresponds to force that moves the bundle laterally toward the stria (i.e., toward the kinocilium in an intact, normal bundle). Note that this definition is different from that used by Kros et al. (2002), where positive driver voltage or displacement indicates excitatory stimuli that open the MET channels, regardless of the direction of the force or bundle movement: response polarity was not explicitly considered due to uncertainty about the orientation of the disorganized hair bundles.”

These statements indicate that they believe that they recorded the “reverse current” and the data is consistent with the “reverse current” since they were likely stimulating in the reverse direction in the Kros et al., 2002 manuscript to elicit the strong inward current responses.

Despite this evidence, we have changed the text to conform with the reviewers wishes:

“Experiments using a *Myo7a* KO mouse model were likely misinterpreted due to the unexpected presence of an additional mechanically-gated current, which in a subsequent study was shown to arise from the PIEZO2 channel^{20,57,58}.”

Lines 439-440: Please, provide reference for the following statement, “Second, a recent study in rats reported that tension in individual tip links is generally lower in IHCs compared to OHCs.” I guess, it is the ref. 55?

Response:

Yes, it is indeed ref. 55. We have added this to the text.

Reviewer #3

Remarks to the Author:

In this second revision, the authors answered all my major comments. Therefore, I do endorse the publication – it's an important and nice work!

However, I still disagree with the authors on the interpretation of the data of Kros et al. (2002). The authors are nitpicking on the ambiguous description of fluid-jet stimulation in Kros et al. (2002). Yet, Kros et al. (2002) did measure the actual bundle movements and, therefore, they should certainly detect "abnormal" transduction to negative bundle deflections. Thus, a cautious researcher would conclude that there is at least a yet unresolved discrepancy between Marcotti et al. (2014) and Kros et al. (2002). If the authors want to show a somewhat uncritical analysis of the literature, it is up to them - it is not relevant to the current manuscript.